# Differential carbonic anhydrase activities control EBV-induced B-cell transformation and lytic cycle reactivation

**Samaresh Malik**[1], **Joyanta Biswas**[1], **Purandar Sarkar**[1], **Subhadeep Nag**[1], **Chandrima Gain**[1¤a], **Shatadru Ghosh Roy**[1¤b], **Bireswar Bhattacharya**[2], **Dipanjan Ghosh**[2], **Abhik Saha**[1] *

**1** Institute of Health Sciences, Presidency University, Kolkata, West Bengal, India, **2** National Institute of Pharmaceutical Education and Research, Kolkata, West Bengal, India

¤a Current address: University of California, Los Angeles, California, United States of America
¤b Current address: Ben-Gurion University of the Negev, Beersheba, Israel
* abhik.dbs@presiuniv.ac.in

**Data Availability Statement:** RNA sequencing data for EBV infection of primary B-lymphocytes and EBV lytic cycle reactivation have been submitted to GEO repository with accession number

## Abstract

Epstein-Barr virus (EBV) contributes to ~1% of all human cancers including several B-cell neoplasms. A characteristic feature of EBV life cycle is its ability to transform metabolically quiescent B-lymphocytes into hyperproliferating B-cell blasts with the establishment of viral latency, while intermittent lytic cycle induction is necessary for the production of progeny virus. Our RNA-Seq analyses of both latently infected naïve B-lymphocytes and transformed B-lymphocytes upon lytic cycle replication indicate a contrasting expression pattern of a membrane-associated carbonic anhydrase isoform CA9, an essential component for maintaining cell acid-base homeostasis. We show that while CA9 expression is transcriptionally activated during latent infection model, lytic cycle replication restrains its expression. Pharmacological inhibition of CA-activity using specific inhibitors retards EBV induced B-cell transformation, inhibits B-cells outgrowth and colony formation ability of transformed B-lymphocytes through lowering the intracellular pH, induction of cell apoptosis and facilitating degradation of CA9 transcripts. Reanalyses of ChIP-Seq data along with utilization of EBNA2 knockout virus, ectopic expression of EBNA2 and sh-RNA mediated knockdown of CA9 expression we further demonstrate that EBNA2 mediated CA9 transcriptional activation is essential for EBV latently infected B-cell survival. In contrast, during lytic cycle reactivation CA9 expression is transcriptionally suppressed by the key EBV lytic cycle transactivator, BZLF1 through its transactivation domain. Overall, our study highlights the dynamic alterations of CA9 expression and its activity in regulating pH homeostasis act as one of the major drivers for EBV induced B-cell transformation and subsequent B-cell lymphomagenesis.

## Author summary

Epstein-Barr virus (EBV) is an enveloped double-stranded DNA tumor virus that is associated with numerous human malignancies, including lymphomas originated from B- and T-cells along with epithelial- and endothelial-derived carcinomas. Alike any other

GSE235941, GSE237484, respectively. All data generated and analyzed in this study are included in the manuscript and supporting files.

**Funding:** This work was supported by grants from DBT/Wellcome Trust India Alliance (IA/I/14/2/501537) and Department of Science and Technology (DST), Govt. of India (CRG/2018/001044) to AS. SM, JB, SN, CG are the recipients of UGC-NET Research Fellowship, respectively, India and PS is recipient of CSIR-NET Research Fellowship. The funders had no role in study design, data collection and analysis, decision to publish, or preparation of the manuscript.

**Competing interests:** The authors have declared that no competing interests exist.

herpesviruses, EBV also displays a biphasic life cycle–latent and lytic replications. While viral latency is associated with different cancer forms, periodic reactivation into lytic cycle replication is important for the production of viral progeny and further transmission to uninfected host cells. Herein, we report that a membrane-associated carbonic anhydrase isoform CA9 plays a key role during the EBV induced B-cell pathogenesis via critically modulating cell acid-base balance. While an elevated CA9 function is essential for EBV induced efficient B-cell transformation and survival of transformed B-lymphocytes, an intermittent switch to lytic replication cycle significantly blocks its activity.

## Introduction

Epstein-Barr virus (EBV) is a ubiquitous human gammaherpesvirus that asymptomatically infects more than 95% of the world's population for the life time of the host [1,2]. EBV has been shown to be causally linked to the development of multiple cancers both of lymphoid and epithelial origin, contributing to ~200,000 cases annually worldwide [2]. EBV preferentially infects B-lymphocytes and develops several B-cell malignancies including endemic Burkitt's lymphoma (BL), Hodgkin lymphoma (HL), diffuse large B-cell lymphoma (DLBCL), and primary central nervous system lymphoma [3,4]. Studies indicate a strong association of EBV-associated malignancies with the impaired function of the host immune system, resulting in post-transplant lymphoproliferative disorders (PTLDs) in immunocompromised individuals, such as patients with HIV infection or patients undergoing immunosuppressive therapies during organ transplantation [1,5]. One of the striking features of EBV life cycle is its capacity to transmute naïve B-lymphocytes into hyperproliferating B-cell blasts followed by the establishment of latency programs (I, II and III) accompanied by different set of expression patterns of viral oncoproteins including six nuclear antigens (EBNAs) and three membrane associated proteins (LMPs) along with several non-coding RNAs[1,2]. EBV utilizes these three distinct latency programs that ultimately direct B-cell blasts to enter germinal centres (GC) and subsequently differentiate into memory B-cells, which serve as the reservoir for lifelong viral infection. *In vitro* stimulation of resting B-cells either by EBV infection or mitogen treatment leads to a transient period of hyperproliferation reminiscent of a GC reaction [6,7]. Increasing evidence suggests several EBV oncoproteins disrupt major B-cell activation pathways, which are normally active in lymph node GC reactions [7]. During initial 2–4 days post-infection, EBV profoundly modifies B-cell architecture, where EBV latent oncoprotein mainly EBNA2 promotes B-cell blasts by transcriptional activation from multiple cellular and viral promoters expressing cell proto-oncogene cMyc and viral EBNA3s and LMPs, which are required for further maintenance of B-cell hyperproliferation [8]. Finally, activated B-cell blasts transform into lymphoblastoid physiology (latency III), where the entire repertoire of EBV latent antigens express, which further alter B-cell physiology including B-cell growth, survival, and GC entry [7,8]. While EBV-positive B-cells that effectively steer the GC reaction exit and develop memory B-cells, non-infected GC B-cells exit and differentiate into plasmablasts [7]. Like any other herpesvirus, EBV also possesses a bi-phasic life cycle including latent and lytic stages. Plasma cell differentiation triggers viral lytic cycle replication, which is critical for the maintenance of viral progeny and cell-to-cell spread as well as transmission from one host to another [9]. EBV encoded immediate early protein BZLF1 represents the master regulator for lytic cycle induction, which eventually triggers expression of ~30 early lytic genes [10]. *In vitro* EBV-mediated growth transformed B-lymphocytes, also known as lymphoblastoid cell lines (LCLs) expressing latency III program, serve as a surrogate model to study the underlying

mechanisms that govern EBV induced B-cell lymphomagenesis [1,8]. To study viral lytic cycle reactivation, LCLs are treated with different agents such as anti-IgG antibodies or combinations of 12-0-tetradecanoylphorbol-13-acetate (TPA) and sodium butyrate [10–12].

Accumulating evidence suggests that cancer cells while transitioning to a hypoxic environment or metabolically shifting towards a rapid aerobic glycolysis state (also known as "Warburg Effect"), there is a considerable decrease in the extracellular pH (pHe; ranges from 6.5–7.1) as compared to the intracellular pH (pHi; $\geq$ 7.2). This unique pH profile allows aberrant cancer cell proliferation through bypassing cell-cycle check points and apoptotic pathways [13–15]. Carbonic anhydrases (CAs; EC 4.2.1.1) belongs to the superfamily of metalloenzymes, which catalyze the production of $H^+$ and HCO3$^-$ ions in the reversible reaction $H_2O + CO_2 = H^+$ + HCO3$^-$ and thereby maintaining the pH homeostasis [16,17]. To date, twelve enzymatically active human CA isozymes have been identified, namely, the cytosolic CA1, CA2, CA3, CA7 and CA13; the mitochondrial CA5A and CA5B; the secretory CA6; and the membrane-associated CA4, CA9, CA12, and CA14 [16]. Studies demonstrated that a cooperative function between cytosolic CA isoforms (CA1 and CA2) and membrane-associated CAs (CA9, CA12 and possibly CA14) is important for pH regulation between the extracellular and intracellular tumor environment [16]. Specifically, the membrane associated CA isoforms convert the excess $CO_2$ present in the extracellular space, to HCO3$^-$ and $H^+$ ions. While the free protons remain outside thereby lowering the pHe, HCO3$^-$ ions are transported back into the cytoplasm and converted to $CO_2$. The $CO_2$ is further utilized in different metabolic pathways or transported back to the extracellular space, thus maintaining the pHi at the normal level [16]. Given the elevated expressions in different cancer types, CA9 and CA12 are considered as tumor-associated CAs, and thus are most studied CA isoforms in respect to tumor settings. However, as compared to CA12, CA9 has gained more attention owing to its significant overexpression in multiple aggressive tumors with poor prognosis as well as association with chemotherapeutic resistance [18–24]. Moreover, *CA9* promoter region contains a single HRE (hypoxia-responsive element) that is essential for its transcriptional activation during hypoxia [25,26].In addition to the fundamental CA activity, CA9 can also influence several other cellular processes [27]. For example, unlike any other CAs, CA9 through its proteoglycan domain contributes to cell adhesion property, thus plays an important role in tumor invasiveness [27]. All these attributes make CA9 as a more attractive therapeutic target than CA12 for drug development.

Increasing evidence suggests that metabolic stress is a major barrier to EBV-induced B-cell transformation, which accounts to extensive remodelling of host metabolic pathways [6,10,28–30]. Despite the critical involvement of CAs, particularly membrane associated CA isoform CA9, in regulating pH, hypoxia, metabolism and tumor development, no reports described potential role of CA9 or per se any CA isoforms in EBV mediated B-cell lymphomagenesis. Our RNA-Seq analyses demonstrated that while CA9 expression was elevated upon EBV infection of naïve B-lymphocytes, its expression was significantly downregulated during viral lytic cycle reactivation. The present work was therefore aimed at exploring CA9 expression and its activity during EBV induced B-cell transformation and subsequent B-cell survival. In sum, our results identify variable CA9 expression and its associated activity during primary infection and lytic cycle replication, which we propose could be exploited for the treatment of multiple EBV-associated B-cell lymphomas.

## Results

### Early EBV infection in naïve B-lymphocytes upregulates CA9 expression

It has been suggested that EBV infection of metabolically dormant B-lymphocytes during early stage of infection results in a transient period of hyperproliferation, resembling a germinal

centre reaction [6]. During this stage, EBV encoded oncoprotein EBNA2 through transcriptional activation of both cell and viral genes radically reprograms B-cell architecture, which in turn transforms into B-cell blasts [8,10,30]. To determine EBV induced global transcriptional alterations during this period, we conducted RNA-Seq analyses of peripheral blood mononuclear cells (PBMCs) from a single donor infected with GFP-tagged EBV (MOI: 10) generated from a stably infected BACmid HEK293T cell line for 2–4 days (Fig 1A and 1B). EBV infection in PBMCs for the indicated time period was checked using immunofluorescence and qPCR analyses (S1 Fig). Across both replicates, 272 and 938 genes were upregulated, while 380 and 1071 genes were significantly downregulated in 2 and 4 day post-infections (DPI), respectively (Fig 1B). Gene ontology analyses demonstrated that as expected EBV infection provoked several known pathways including cell division, DNA replication, cell proliferation and glycolytic process, while repressed inflammatory and immune response signalling cascades (Fig 1C). Among the upregulated pathways, we further investigated the expression patterns of 'cellular response to hypoxia' target genes during EBV induced B-cell transformation (Figs 1C and 2A and 2B). Previous studies suggest that EBV infection in contrast to mitogen induced B-cell activation leads to stabilization of hypoxia-induced factor 1 alpha (Hif1$\alpha$), required for aerobic glycolysis [31]. Although Hif1$\alpha$ transcripts were present at higher level, our RNA-Seq data indicated Hif2$\alpha$ (or EPAS1) instead was significantly transcriptionally activated in EBV induced B-cell blasts as compared to the resting B-lymphocytes (Fig 2A).

CA9, a membrane associated CA isoform, is known to be induced by hypoxia, involved in adaptation to acidosis and implicated in cancer progression via its catalytic and/or non-catalytic functions [16,24,32]. Although CA12 is not regulated by hypoxia, along with CA9, CA12 represents another membrane associated CA implicated in cancer development [16,33,34]. Out of 15 CA isoforms, CA9 and to a lesser extent CA12 were significantly elevated upon EBV infection in primary B-lymphocytes (Fig 2B). Since RNA-Seq was performed in single donor of PBMCs, to nullify genetic discrepancies the data was further validated using two different donors in a similar experimental set up (Fig 2C–2E). Quantitative real time PCR, western blotting and immunofluorescence studies demonstrated expressions of both CA9 and CA12 were significantly elevated in response to EBV infection in naïve B-lymphocytes (Fig 2C–2E). It is known that mitogens including CpG oligo TLR9 ligand ODN2006 can promote B-cell proliferation as similar to EBV infection [35]. Congruent with EBV infection, mitogen induced B-cell activation similarly transcriptionally activated CA9 and CA12 expressions along with hypoxia responsive factors (S2 Fig), indicating the critical involvement of both hypoxia and CA-mediated pH regulation during early stage of B-cell activation, irrespective of viral infection.

## EBV transformed B-lymphocytes express elevated levels of two membrane associated CA isoforms–CA9 and CA12

EBV induced transcriptional activation of CA9 and CA12 expressions during early stage of infection of naïve B-lymphocytes prompted us to further profile CA expressions in transformed B-lymphocytes. qPCR analyses of two LCLs–LCL#1 and LCL#89 demonstrated significant upregulation of only CA9 and CA12 transcripts when compared to resting B-lymphocytes (Fig 3A and 3B). As similar to primary infection model, significant transcriptional activation of EPAS1, but not Hif1$\alpha$, was observed in both LCLs (Fig 3A and 3B). The transcriptional activation of CA9, CA12 and EPAS1 in LCLs was further validated by western blot analyses (Figs 3C and S3). Next, the transcripts profile of every member of CA and Hif family in LCLs were corroborated using the RNA-Seq data of 'Genotype-Tissue Expression (GTEx; https://gtexportal.org/home/) project (Fig 3D). The results demonstrated that while CA2 and CA12 were downregulated, CA9 expression was significantly upregulated in EBV

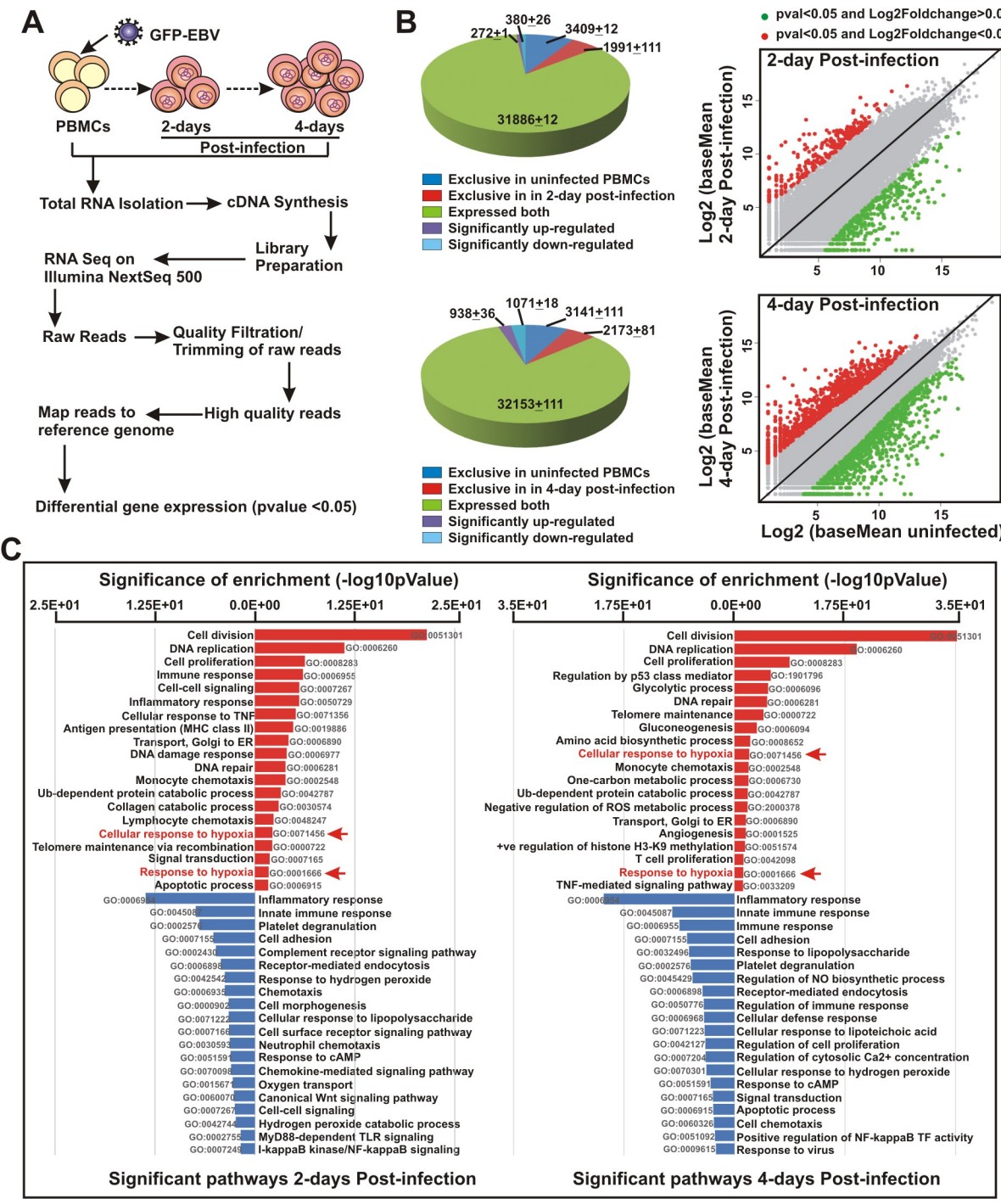

**Fig 1. Whole transcriptome analysis of EBV infected resting B-cells during initial phase of infection.** (A) Schematic representation of whole transcriptome analysis of PBMCs infected with GFP-EBV for 0, 2 and 4 days post-infection (DPI) in Illumina platform as described in the "Materials and Methods" section. (B) Pie chart and scattered plot analyses of differentially expressed gene sets in 2 and 4 DPI. (C) Differentially expressed gene sets were uploaded on DAVID v6.8 webserver for functional analysis. Gene Ontology (GO) was selected from the hits table for DAVID clustering. The bar diagrams (upregulated: red; downregulated: blue) represent top 20 most significantly affected pathways.

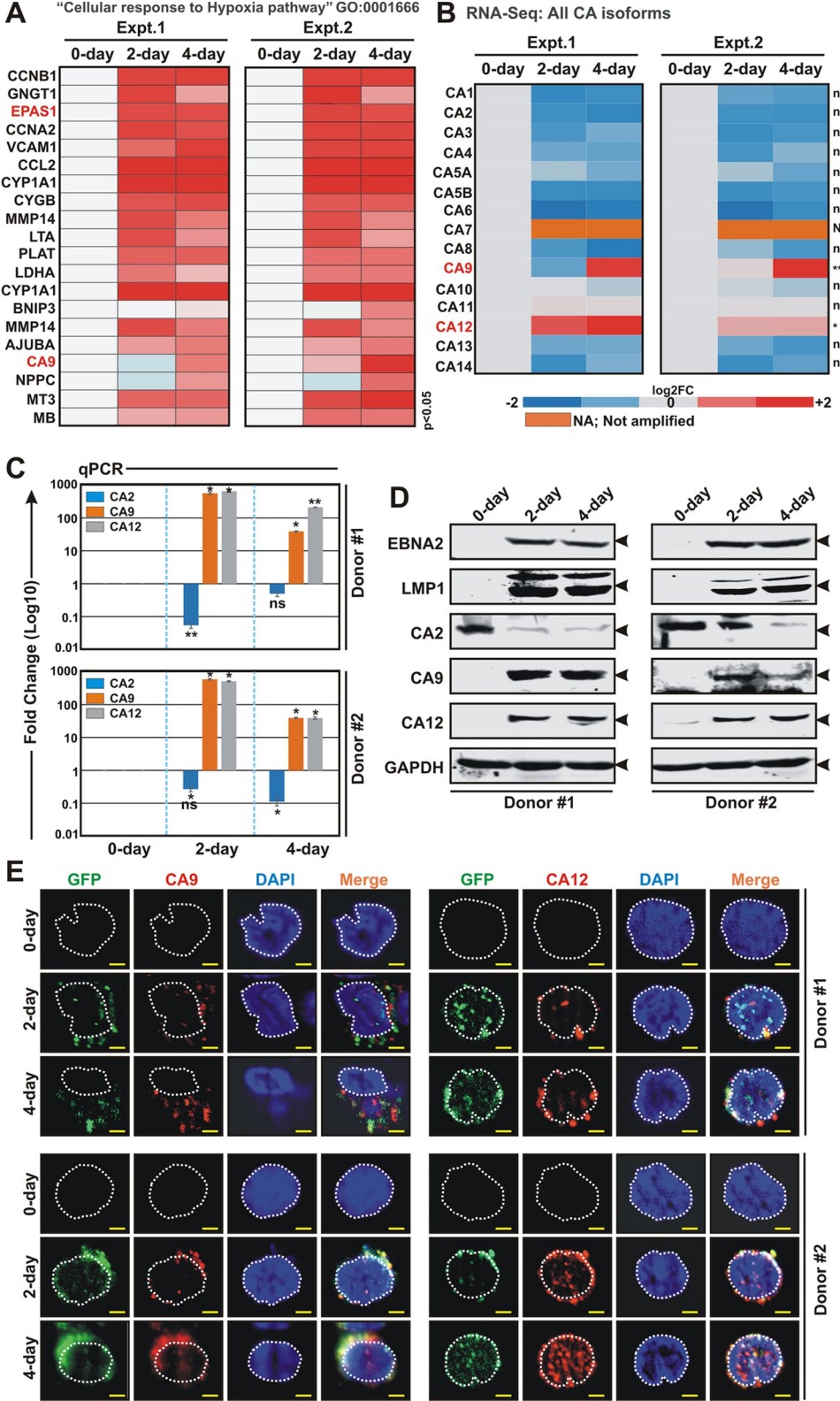

**Fig 2. EBV infection of naïve B-lymphocytes specifically enhances expressions of two membrane associated carbonic anhydrase isoforms—CA9 and CA12.** (A) Heat map analysis of differentially expressed genes under "Cellular response to Hypoxia pathway" GO: 0001666 from RNA-Seq analyses of PBMCs infected with GFP-EBV.

Differential gene expressions were performed based on p-value as < 0.05 and log2 Fold Change as 2 and above (upregulated, red) and -2 and below (downregulated, blue). (B) Heat map analysis (log2 Fold Change) of all carbonic anhydrase (CA) isoforms in RNA-Seq analyses of PBMCs infected with GFP-EBV for 0, 2 and 4 days post-infection (DPI). (C-E) PBMCs from two individual donors were infected with GFP-EBV for the indicated time points and subjected to (C) quantitative real-time PCR (qPCR), (D) western blot and (E) immunofluorescence analyses. (C) The relative changes in transcripts (log10) of the selected genes using the $2^{-\Delta\Delta Ct}$ method are represented as bar diagrams in comparison to uninfected PBMC-control using B2M as housekeeping gene. Two independent experiments were carried out in similar settings and results represent as an average value for each transcript. *, **, *** = p-value < 0.01, 0.005 and 0.001 respectively. (D) 0, 2 and 4 days post-infected cells were harvested, washed with 1 x PBS, lysed in RIPA buffer and subjected for western blot analyses with the indicated antibodies. GAPDH blot was used as loading control. Representative gel pictures are shown of at least two independent experiments. (E) For immunofluorescence studies, ~5 x $10^4$ post-infected cells at the indicated time points were fixed with 4% paraformaldehyde and incubated with the indicated primary antibodies followed by Alexa Fluor conjugated secondary antibodies for visualization in a Zeiss fluorescence microscope with Apotome attachment. Nuclei were counterstained using DAPI (4',6'-diamidino-2-phenylindole) before mounting the cells. All panels are representative pictures of two independent experiments. Scale bars, 5 μm.

transformed LCLs as compared to whole blood samples (Fig 3D). Similarly, reanalyses of microarray data of EBV infected BL31 cell line [36] also indicated significant upregulation of CA9, but not CA12 expression as compared to the uninfected control cells (S4 Fig). These analyses combined with our data from primary infection and transformed B-lymphocytes models indicated CA9 might be the only prominent membrane associated CA isoform expressed in B-lymphocytes in response to EBV infection. In order to confirm this notion, LCLs were fractionated into different cellular compartments using centrifugation method (Fig 3E). Measurements of CA activities of equivalent protein amount of whole cell lysate (WCL), cytoplasmic, membrane, nuclear and mitochondrial fractions demonstrated CA activity was only associated with membrane fraction along with WCL (Fig 3D), suggestive of significant presence of membrane associated CA isoform(s), at least CA9, in EBV transformed B-lymphocytes.

## Carbonic anhydrase activity is required for EBV induced B-cell transformation and transformed B-cells outgrowth

Given their essential roles in regulating pH balance and tumor association, substantial efforts have been made to generate specific CA inhibitors (CAi) against two membrane associated CA isoforms—CA9 and CA12 as anticancer therapy for the last two decades [37,38]. Owing to its limited expression in normal cells as well as responsiveness in hypoxic tumor microenvironment, CA9 was proposed to be more druggable candidate as compared to CA12 [37]. Given the elevated expression pattern of CA9 and to a lesser extent CA12 in response to EBV infection, we first checked cell viability of LCLs in the absence and presence of two CA9/CA12 specific inhibitors–SLC-0111/U-104 and compound S4 along with pan CA inhibitor acetazolamide (Figs 4 and S5 and S1 Table). While SLC-0111/U-104 is currently under Phase Ib/II clinical trials as adjuvant agent in advanced solid tumors [39], efficacy of compound S4 is now being evaluated in preclinical studies [40]. Acetazolamide, a first generation CA inhibitor, is used to treat metabolic alkalosis [41]. Inhibition of CA activity with increasing concentrations of all three inhibitors in both the LCLs resulted in lowering of the intracellular pH, decreasing cell viability as well as colony formation ability in a dose-dependent fashion (Figs 4A and 4B and S5). Based on 50% inhibitory concentration ($IC_{50}$) for cell viability (Figs 4A and 4B and S5) as well as inhibitor constant (Ki) (S1 Table), we utilized compound S4 for most of the following studies.

Next, to determine whether inhibition of CA activity would hamper EBV induced B-cell transformation, freshly isolated PBMCs were infected with GFP-EBV (MOI: 10) in the absence

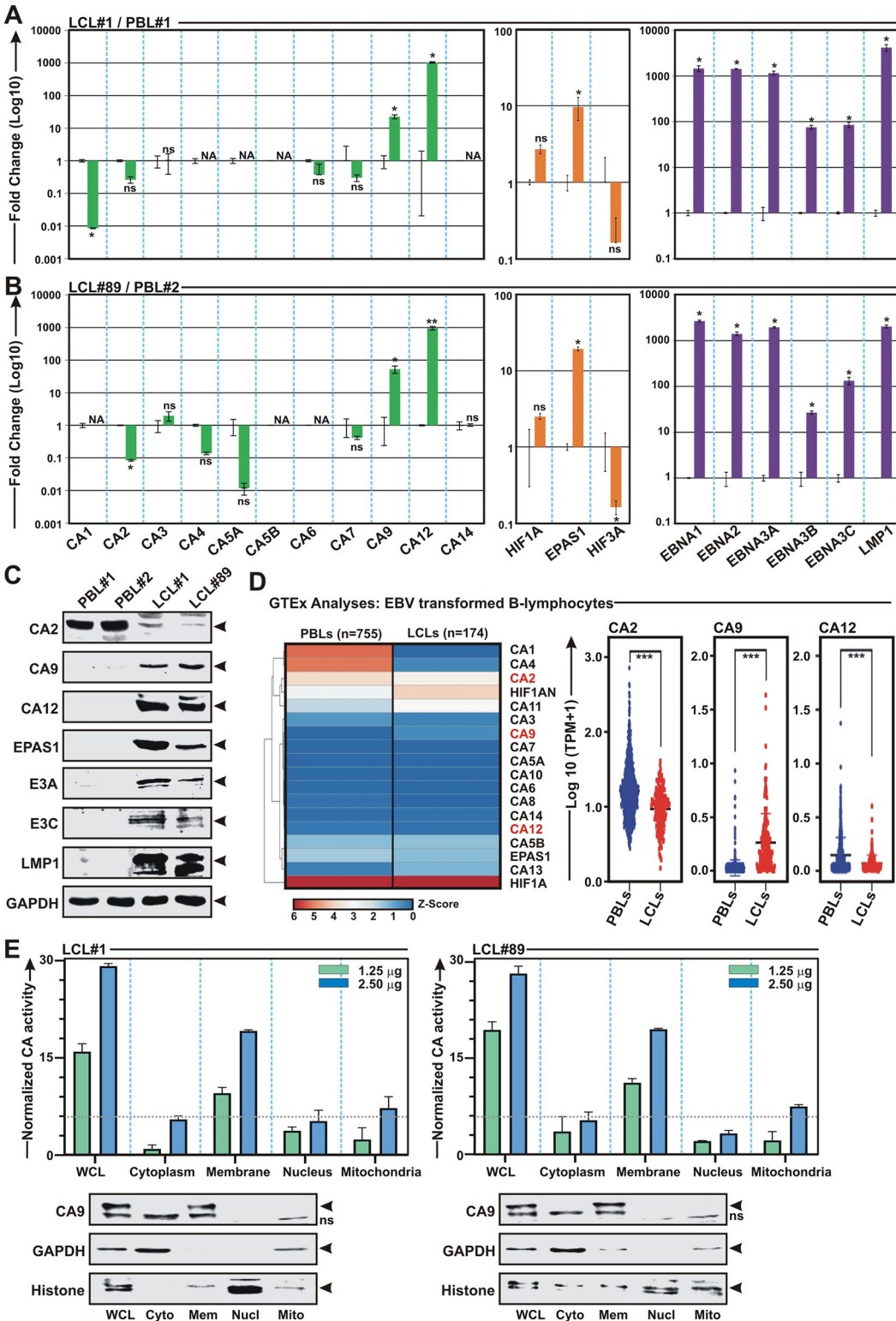

**Fig 3. Elevated levels of CA9 and CA12 expressions are detected in EBV transformed B-lymphocytes.** (A-B) Total RNA or (C) total proteins isolated from two LCL clones–LCL#1 and LCL#89 along with PBMCs from two different donors were subjected for either (A) qPCR or (B) western blot analyses. (A-B) The relative changes in transcripts (log10) of the selected genes using the $2^{-\Delta\Delta Ct}$ method are represented as bar diagrams in comparison to uninfected PBMC-control using B2M as housekeeping gene. Two independent experiments were carried out in similar settings and results represent as an average

value for each transcript. *, **, *** = p-value < 0.01, 0.005 and 0.001 respectively. (C) For Western blot analyses, ~ 5 million cells were harvested, washed with 1 x PBS, lysed in RIPA and fractionated using appropriate SDS-PAGE followed by incubation with the indicated antibodies. GAPDH blot was performed as loading control. Representative gel pictures are shown of at least two independent experiments. (D) Heat map and dot plot analysis of the transcripts profile of the indicated genes in whole blood cells and LCLs using 'Genotype-Tissue Expression (GTEx)' project (https://gtexportal.org/home/). (E) ~ 20 million LCLs were fractionated into nuclear, membrane, mitochondrial and cytoplasmic cell fractions using centrifugation methods, followed by carbonic anhydrase assay according to the manufacturer's instructions. Bar diagrams are the average of two independent experiments. A small portions were subjected to western blot analyses with the indicated antibodies. GAPDH and histone blots were used as reference proteins for cytoplasmic and nuclear fractions, respectively.

and presence of CAis for 28 days (Figs 5A and 5B and S6A). To nullify any off-target effects we utilized two CAis–compound S4 and U-104 and subsequently treated EBV infected PBMCs (Figs 5A and 5B and S6A, respectively). Fluorescence microscopy analyses at each time points (0, 2, 4, 7, 14, 21 and 28 DPI) demonstrated that treatment with both compound S4 and U-104 significantly retarded viral B-cell transformation ability in all three PBMC donors (Figs 5A and 5B and S6A). Whether stunted B-cell transformation in the presence of CAi was due to the inhibition of CA catalytic activity and subsequent pH imbalance, a fraction of compound S4 treated cells at each time point was tested for CA activity and intracellular pH (Fig 5C and Fig 5D, respectively). In contrast to DMSO control, compound S4 treatment caused a significant decrease in CA activity and intracellular pH during EBV induced B-cell transformation (Fig 5C and Fig 5D, respectively). Notably, in control experiment, CA activity was immediately increased 2 DPI, peaked 4 DPI, followed by a slight decreased at 7 DPI and maintained over the B-cell transformation time course (Fig 5C). Congruent with CA activity, qPCR analyses also demonstrated similar temporal expression pattern of CA9 transcripts during EBV induced B-cell transformation, while both compound S4 and U-104 treatment resulted in significant reduction (~10–50 fold) of CA9 transcripts as early as 2 DPI (Figs 5E and S6B). Analogous CA9 expression profiles were also observed in reanalyses of two different RNA-Seq of similar time point data sets (GSE125974 and DRA011328) during EBV infection of primary B-lymphocytes [42,43] (Fig 5F). Both our qPCR and RNA-Seq reanalyses data demonstrated that CA9 expression peaked 2–4 DPI and slowly decreased afterwards (Fig 5E and 5F). In addition, reanalyses of both RNA-Seq data revealed that among all the CA isoforms, only CA9 typically expressed 2–4 DPI during EBV induced B-cell transformation (S6C Fig). Collectively, these data suggest that EBV infection alters CA9 expression and its activity during the early infection stages to support the significant physiological shift from quiescence to rapid lymphoblastic proliferation.

## Carbonic anhydrase inhibitor causes apoptotic cell death in EBV transformed B-lymphocytes

Next, we wanted to evaluate the underlying mechanism(s) that governs CAi-mediated cell death in LCLs and during B-cell transformation events. Studies suggest that CAis induce cell death in multiple cancer cell lines through apoptotic induction [44–46]. To determine whether CA activity inhibition results in cell apoptosis in LCLs, cells were either left untreated (DMSO control) or treated with 0.1 mM CAi compound S4 for 24h and subjected to FITC-labelled annexin V and propidium iodide (PI) double staining (Fig 6A and 6B). While early apoptosis is indicated by only green staining (FITC) in the plasma membrane, late apoptosis lacking membrane integrity is observed by red staining (PI) throughout the nucleus and a halo of green staining (FITC) on the cell surface. Both fluorescence microscopy and flow cytometry analyses demonstrated that LCLs treated with CAi S4 was significantly enriched by double staining, signifying late apoptosis (Fig 6A and Fig 6B, respectively). Western blot and qPCR

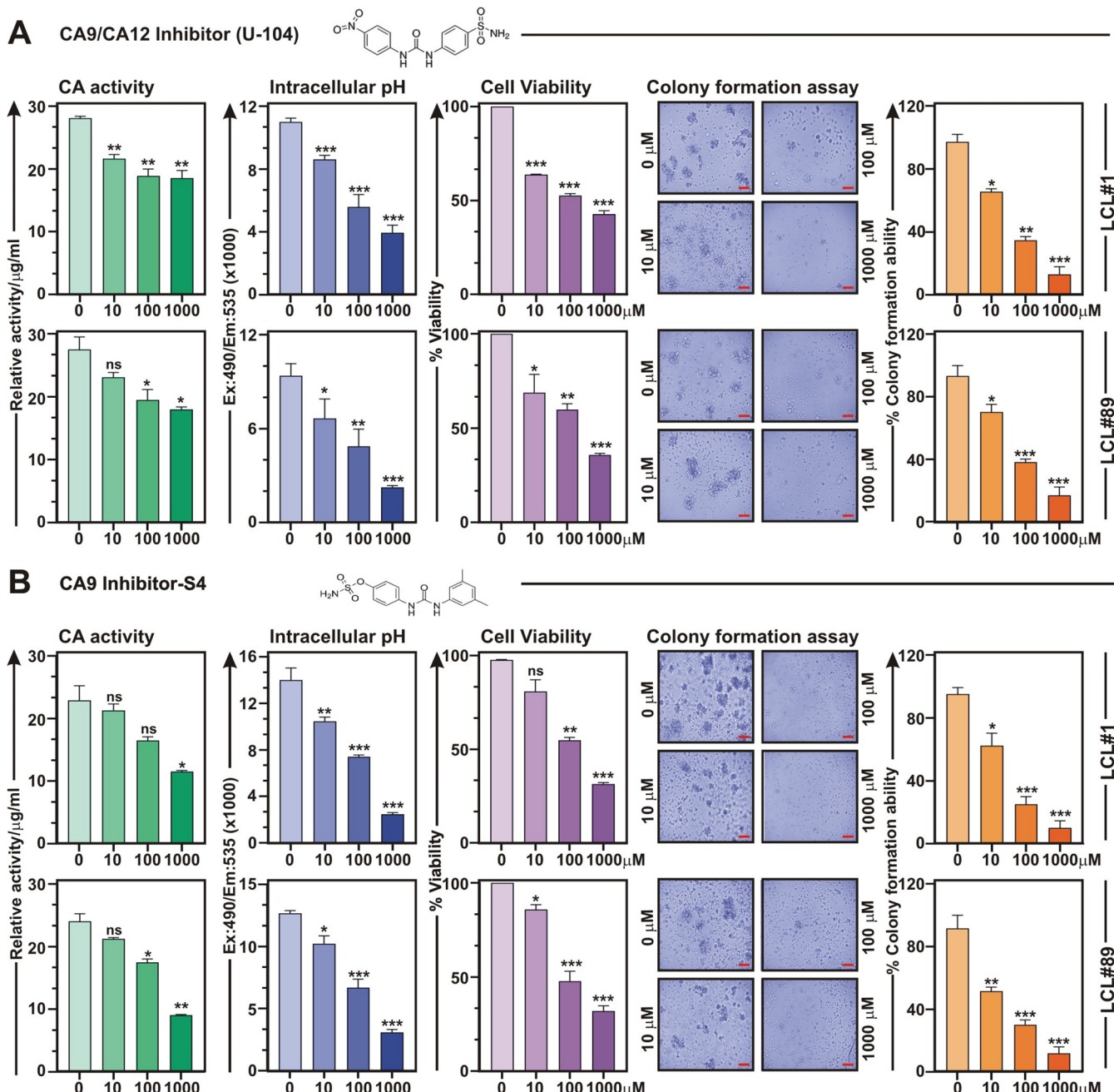

**Fig 4. Carbonic anhydrase inhibitors induce cell death in EBV transformed B-lymphocytes.** (A-B) LCLs (LCL#1 and LCL#89) were either left untreated (DMSO control) or treated with increasing concentrations (10–1000 μM) of CA9/CA12 specific inhibitors—(A) SLC-0111/U-104 or (B) S4 for 24 h and measured carbonic anhydrase activity, intracellular pH, cell viability and colony formation ability as described in the "Materials and Methods section". Carbonic anhydrase activity and intracellular pH determination assays were performed using kits as per manufacturer's protocols. For cell viability assays, viable cells from each well of 6-well plates were measured by Trypan blue exclusion method using an automated cell counter. For the colony formation assay, 14 days post-treatment colonies on the soft agar were photographed (bright-field) using a Fluorescent Cell Imager. Scale bars, 100 μm. The number of colonies were measured by ImageJ2 software and plotted as bar diagrams. Error bars represent standard deviations of duplicate assays of two independent experiments. *, **, *** = p-value < 0.01, 0.005 and 0.001 respectively.

analyses further showed drastic elevation of p53 tumor suppressor and its downstream regulator CDKN1A/p21$^{Waf1/Cip1}$ expressions at both protein and transcripts levels (Fig 6C and Fig 6D, respectively), indicative of cell-cycle arrest and apoptosis. PARP cleavage in western

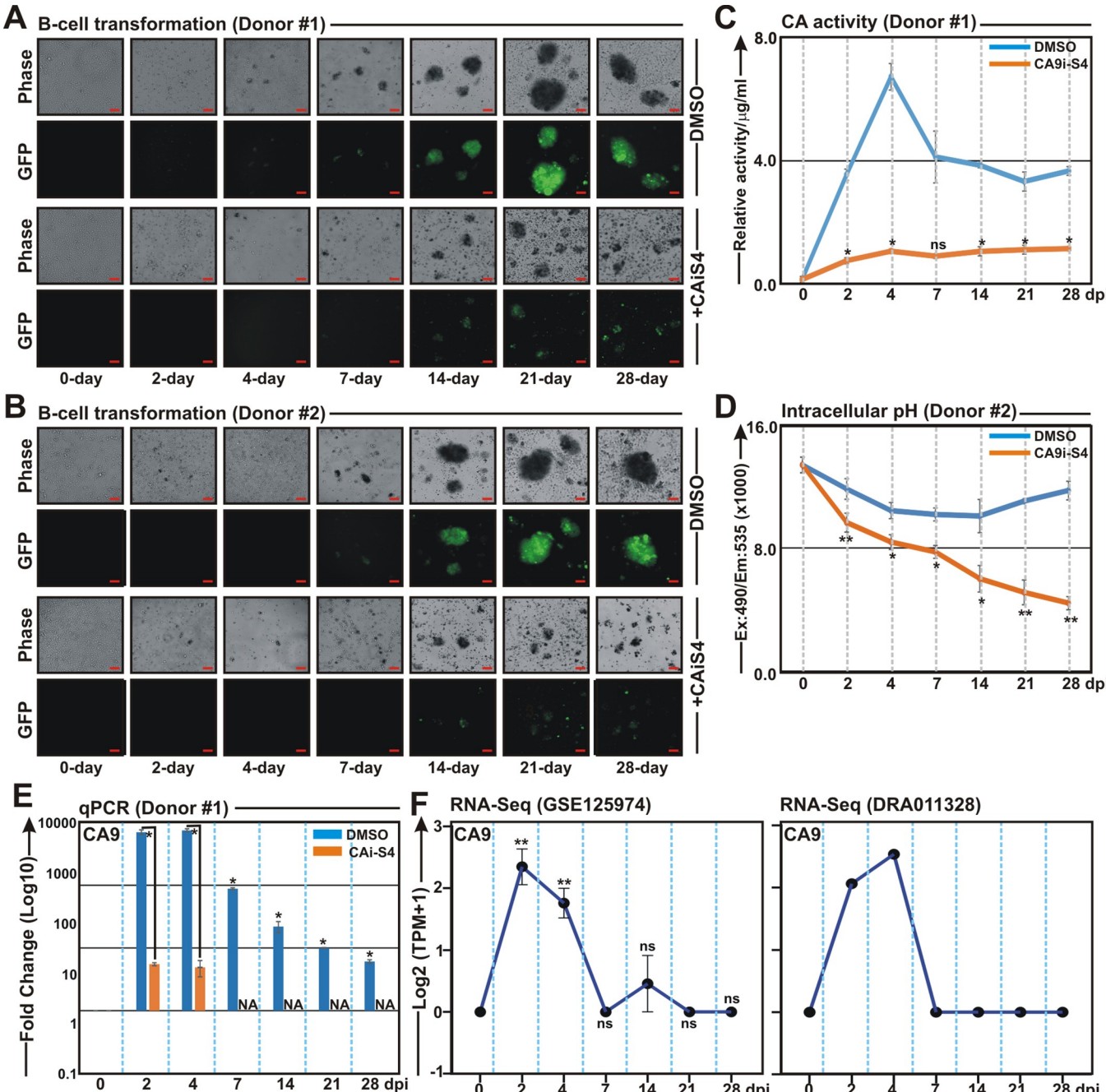

**Fig 5. Carbonic anhydrase inhibitor retards EBV induced B-cell transformation.** (A-B) PBMCs isolated from two individual donors (donor #1 and donor #2) were infected with GFP-EBV (MOI ~10) in the absence (DMSO control) or in the presence of 0.1 mM carbonic anhydrase inhibitor S4 for 28 days. At the indicated time points post-infected cells were photographed using a Fluorescent Cell Imager. Scale bars, 100 μm. (C-D) In a similar experimental set up, cells at the indicated time points were harvested and subjected to either (C) carbonic anhydrase activity assay, (D) intracellular pH detection and (E) qPCR analyses. (C) Carbonic anhydrase activity and (D) intracellular pH determination assays were performed using kits as per manufacturer's protocols. (E) The relative changes in transcripts (log10) of the selected genes using the $2^{-\Delta\Delta Ct}$ method are represented as bar diagrams in comparison to uninfected PBMC-control using B2M as housekeeping gene. (C-E) Average values +/- SEM are plotted of two independent experiments performed in similar settings. *, **, *** = p-value < 0.01, 0.005 and 0.001 respectively. (F) Reanalyses of RNA-Seq data (GSE125974 and DRA011328) of B-cells infected with EBV for the indicated time points showing temporal expression prolife of CA9 transcript.

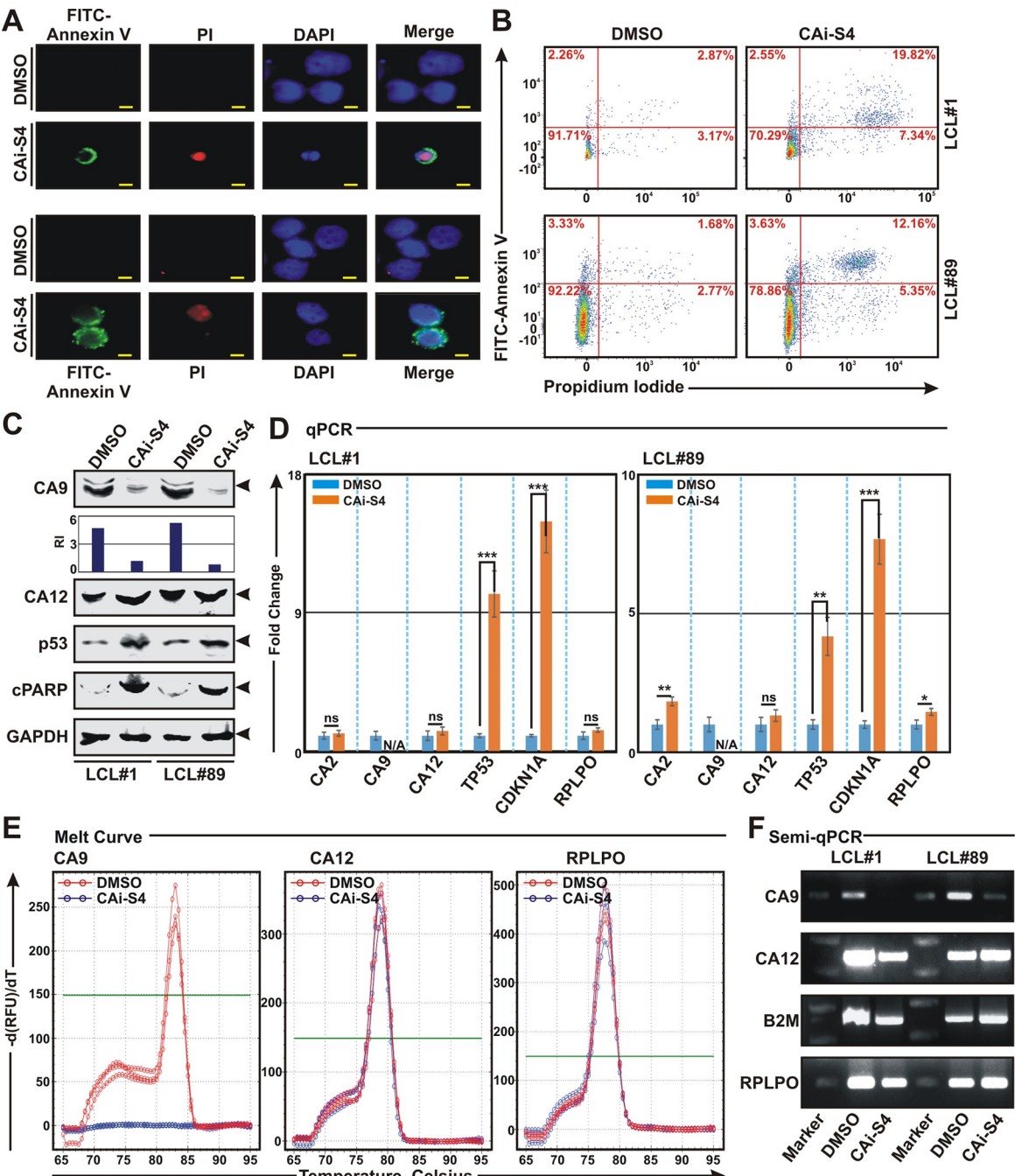

**Fig 6. Carbonic anhydrase inhibitor causes cell apoptosis in EBV transformed cells.** (A-F) ~ 10 million LCLs (LCL#1 and LCL#89) were either left untreated (DMSO control) or treated with 0.1 mM carbonic anhydrase inhibitor S4 for 24 h and subjected to (A-B) apoptosis assay using annexin V/propidium iodide (PI) staining, (C) western blot analyses using the indicated antibodies, (D-E) real-time qPCR analyses for the indicated genes and (F) semi-quantitative PCR analyses as described in the "Materials and Methods" section. (A-B) Representative images of (A) fluorescence microscopy and (B) flow cytometry analyses showing cell apoptosis in LCLs after 0.1 mM S4 treatment. (A) DAPI stained cell nucleus, FITC-labelled Annexin V and PI were visualized by blue (Ex = 359 nm; Em = 457 nm), green (Ex = 488 nm; Em = 350 nm) and red (Ex = 586 nm; Em = 603 nm) signals, respectively using a Ziess fluorescence microscope. Scale bars, 10 μm. (B) FITC labelled annexin V binding and PI staining were detected using a BD cell analyzer. Results were analysed using Floreada.io, an online based tool for flow cytometry data analyses. (C) Representative western blot images of two independent experiments. GAPDH blot was used as loading control. Protein bands were quantified by Odyssey imager software and indicated as bar diagrams at the bottom of corresponding lanes. (D) The relative changes in transcripts of the selected genes using the $2^{-\Delta\Delta Ct}$ method are represented as bar diagrams in comparison to DMSO treated samples using B2M as housekeeping gene. Error bars represent standard deviations of duplicate assays of two independent experiments. *, **, *** = p-value < 0.01, 0.005 and 0.001 respectively. (E)

Representative images of melt curve analyses of real-time PCR amplifications of the indicated genes in (D). (F) Representative ethidium bromide stained 2.5% agarose gel images of PCR amplification of the indicated genes using the similar primer sets used in qPCR amplifications in (D).

blot analyses additionally validated apoptotic induction by CAi S4 treatment in LCLs (Fig 6C). Overall, these data suggest that CAi compound S4 induces apoptosis in EBV transformed B-lymphocytes through transcriptional activation of p53 and its downstream regulators.

Previously it has been shown that CA inhibitor SLC-0111/U-104 downregulates CA9 expression at both protein and transcript levels in hepatoblastoma cell lines [47]. Similarly, following compound S4 treatment, levels of CA9 protein and mRNA were significantly decreased in both LCLs (Fig 6C and 6D). Melt curve analyses demonstrated that while CA12 and control RPLPO transcripts remained invariant, no amplification of CA9 transcript was observed after compound S4 treatment (Fig 6E). The qPCR results were further corroborated by semi quantitative PCR in both LCLs with or without compound S4 treatment (Fig 6F).

## EBV oncoprotein EBNA2 and B-cell specific transcription factors are recruited on to distal CA9 promoter/enhancer region

To understand the underlying molecular mechanism governing CA9 transcriptional activation, we reanalysed ChIP-Seq data for all six EBNA proteins—EBNA1, EBNA2, EBNALP, EBNA3A, EBNA3B and EBNA3C along with several B-cell specific transcription factors in ENCODE GM12878 and IB4 LCLs (Figs 7A and S7A and S2 Table). LMP1 constitutively activates NF-κB to promote LCLs growth and survival [48,49]. We therefore included ChIP-Seq data for NF-κB subunits RelA, RelB, cRel, p50, and p52 in our analyses (Fig 7A and S2 Table). Reanalysis of ChIP-Seq data revealed that among all the viral genes only EBNA2 peak in all the ChIP-Seq data (GSE29498, GSE176232, GSE76869) was significantly enriched onto CA9 promoter/enhancer region in both GM12878 and IB4 LCLs (Figs 7A and S7A). EBNA2 peak also converged with several B-cell specific transcription factors including RBP-Jκ (Recombination signal Binding Protein for immunoglobulin Kappa J region), EBF1 (Early B-cell Factor 1), MEF2B (Myocyte Enhancer Factor 2B), IKZF1 (IKAROS Family Zinc Finger 1), EP300 (E1A Binding Protein P300) and SPI1/PU.1 (Fig 7A and S2 Table). Reanalyses of ATAC-Seq data in LCL GM12878 also demonstrated chromatin accessibility in the EBNA2 binding region (Fig 7A and S2 Table), indicating a plausible transcriptional regulation of CA9 expression at the distal promoter/ enhancer region via recruiting distinct B-cell specific transcription factors. However, the ATAC-Seq signal on the EBNA2 binding region was markedly lower compared to the upstream region closer to the ARHGEF39 (Rho Guanine Nucleotide Exchange Factor 39) gene locus (S7B Fig), indicating that ARHGEF39 gene might strongly be regulated than CA9 gene in LCL background. In addition, reanalyses of RNA pol II ChIA-PET and ChIP-Seq data showed both EBNA2 binding upstream and downstream regions were linked by chromosome looping for transcriptional regulation of CA9 and ARHGEF39 (S7B Fig). Reanalyses of RNA-Seq data of PBMCs and GM12878 LCLs indeed demonstrated that a similar phenomenon of higher ARHGEF39 expression (~2 fold) as compared to CA9 expression (S7C Fig).

Among the enriched transcription factors, previous genome wide analyses demonstrated RBP-Jκ had highest co-occupancy with EBNA2 binding regions [50]. EBNA2-associated RBP-Jκ sites had approximately two-fold higher signals than RBP-Jκ sites that lacked EBNA2, indicating a strong EBNA2 effect on RBP-Jκ association with DNA in EBV transformed B-lymphocytes [50]. To examine whether EBNA2 and RBP-Jκ interaction is necessary for CA9 transcription, we analyzed the presence of putative RBP-Jκ binding sites in the CA9 promoter/ enhancer region by employing publicly available transcription factor binding prediction

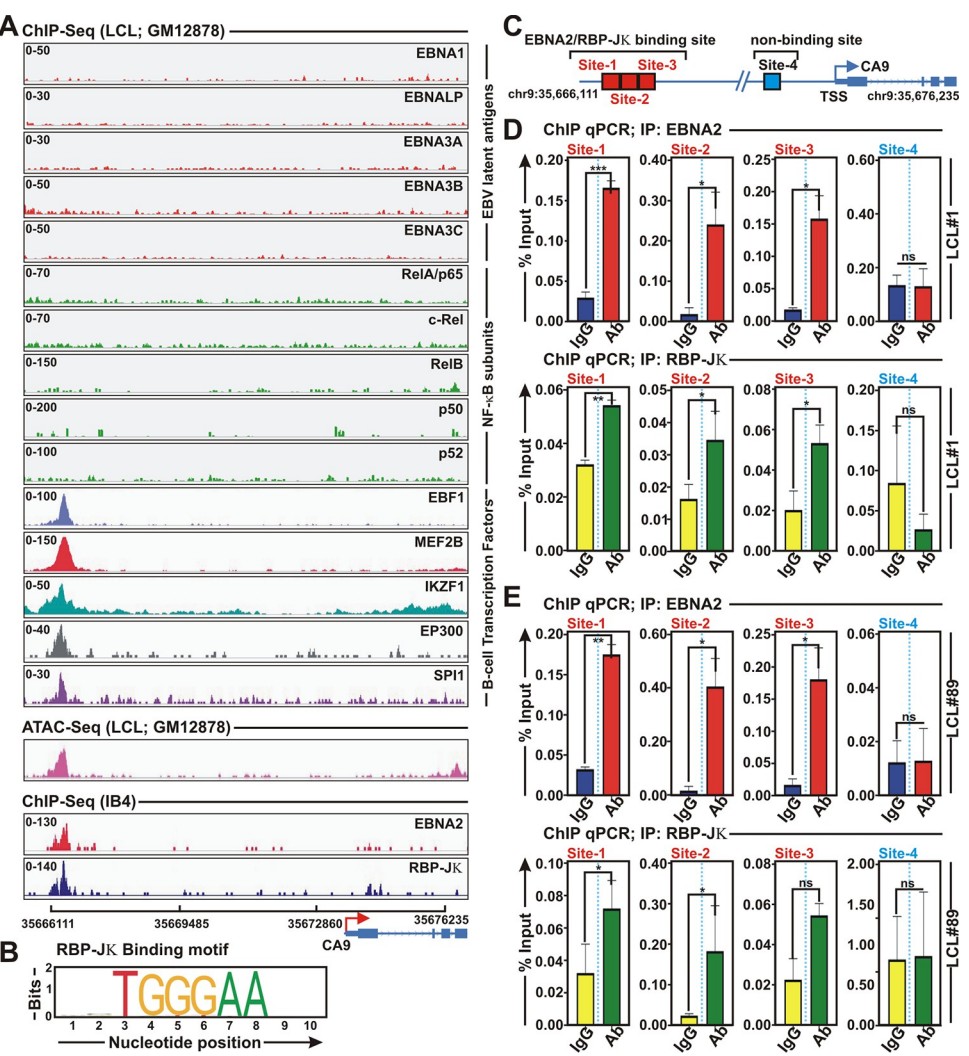

**Fig 7. EBNA2 and several B-cell specific transcriptional factors colocalize onto CA9 promoter region.** (A) ChIP-seq data for the indicated viral antigens and B-cell specific transcription factors were reanalysed and displayed using IGV (Integrative Genomics Viewer) software for CA9 gene locus. (B-C) RBP-Jκ binding motif obtained from JASPAR database in the EBNA2 binding region of CA9 promoter. (C) Cartoon representation of genomic location for three RBP-Jκ binding and non-binding sites on the CA9 promoter region for ChIP-qPCR primer designing. Transcription start site (TSS) is indicated by red arrow. (D-E) ChIP-qPCR showing recruitment of (D) EBNA2 and (E) RBP-Jκ on CA9 promoter region. Site 1, 2 and 3 are the binding region and site 4 is non-binding region. ChIP-qPCR primers were designed by NCBI primer BLAST application. Two independent experiments were carried out in similar settings and results represent as an average value for each genomic segment. *, **, *** = p-value < 0.01, 0.005 and 0.001 respectively.

software, JASPAR (http://www.jaspar.genereg.net). Three distinct putative RBP-Jκ binding motifs were identified in the distal EBNA2 binding region (Fig 7B). To further confirm co-occupancy of EBNA2 and RBP-Jκ on the CA9 promoter region, we designed three primer sets for the distal binding and one primer set for the proximal non-binding regions of CA9 as depicted in the cartoon diagram to perform ChIP-qPCR in two LCLs- LCL#1 and LCL#89 (Fig 7C). Overall, the results demonstrated that both EBNA2 and RBP-Jκ were significantly enriched at all three distal binding sites, while no binding was observed in the proximal non-binding region of CA9 promoter in both LCLs (Fig 7D and Fig 7E, respectively).

## EBNA2 mediated CA9 expression is essential for B-cell survival

EBNA2 recruitment onto CA9 promoter/ enhancer region further prompted us to examine whether loss of EBNA2 gene expression abrogates CA9 expression and its activities during EBV initial phase of infection of primary B-lymphocytes. To this end, we isolated viruses from two EBV positive Burkitt's lymphoma lines—P3HR1 and jiyoye and subsequently infected PBMCs for 2 and 4 days (Fig 8A). While P3HR1 contains type 2 EBV strain with naturally occurring EBNA2 deletion, Jiyoye harbours wild-type type 2 EBV strain [51,52]. Owing to EBNA2 deletion, P3HR1 virus unable to transform naïve B-lymphocytes *in vitro* [52] and compared to wild-type B95.8 type 1 strain, P3HR1 virus develops B-cell lymphomas less efficiently in a subset of infected cord blood-humanized mice [53,54]. qPCR analyses of PBMCs infected with either P3HR1 (ΔEBNA2) or Jiyoye (wild-type) virus demonstrated that EBNA2 expression was directly correlated with CA9 but not with CA12 expressions (Fig 8B). However, no significant change was observed for ARHGEF39 gene, upstream of EBNA2 binding region in the CA9 promoter/enhancer region in response to either ΔEBNA2 or wild-type EBV infection of naïve B-lymphocytes (S7D Fig), indicating that EBNA2 might only be involved in downstream CA9 regulation in that genomic region. Elevated CA9 expression was further correlated with higher CA activity and unaltered intracellular pH in wild-type virus infected PBMCs when compared with ΔEBNA2 virus infected PBMCs (Fig 8C and 8D), indicating that EBNA2 mediated CA9 expression plays an important role during early stage of EBV induced B-cell activation. Western blot analyses of P3HR1 and Jiyoye cells evidently demonstrated that EBNA2 expression was directly interrelated with elevated CA9 expressions, while CA12 expressions were unaltered in these two lines (Fig 8E). In addition, no significant correlation was also observed for ARHGEF39 expressions with EBNA2 in these two cell lines (S7E Fig).

A more direct correlation between EBNA2 and CA9 expressions was also validated using P3HR1 null EBNA2 cell line transiently transfected with either vector control or EBNA2 expressing plasmid (Fig 8F). The results demonstrated that EBNA2 expression stimulated CA9 expression, which was typically absent in P3HR1 cells (Fig 8F). ChIP-qPCR analyses also showed both EBNA2 and RBP-Jκ were significantly enriched at all three EBAN2 distal binding sites as compared to the proximal non-binding region of CA9 promoter/enhancer region in EBNA2 expressing P3HR1 cells (S8A–S8C Fig). However, in contrast, no binding of both EBNA2 and RBP-Jκ were observed in vector transfected P3HR1 cells (S8A–S8C Fig). Additionally, significant CA9 expressions were also observed in both EBV positive and high EBNA2 expressing EBV positive B-cell lymphoma lines as compared to EBV negative and low EBNA2 expressing EBV positive B-cell lymphoma lines in DepMap portal (S7F Fig). However, in contrast, no significant change was observed for ARHGEF39 expressions in both groups (S7F Fig), indicating that EBNA2 might only be involved in transcriptional regulation for the downstream target gene CA9 expression in EBV positive B-cells.

'Real-time qPCR analyses showed only mitochondrial CA5A and membrane associated CA9 were significantly elevated in Jiyoye cells as compared to P3HR1 cells, which was further correlated with overall increased CA activity in Jiyoye cells (Fig 8G and 8H). Increased CA activity in whole cell lysates as well as in the membrane and mitochondrial fractions further supported the notion (Fig 8H and 8I). Whether the increased CA activity was specifically due to elevated CA9 expression or combined effect, cells were either left untreated (DMSO control) or treated with 100 µM CA9/CA12 specific inhibitor compound S4 for 24 h and checked for CA activity (Fig 8H). While compound S4 treatment resulted in significant reduction of CA activity in Jiyoye cells, little or non-significant effect was observed for P3HR1 cells (Fig 8H). Cell viability analyses demonstrated that as compared to P3HR1 cells, Jiyoye cells with elevated CA9 expression were more sensitive towards cell death induced by CAi S4 in a dose dependent fashion (Fig 8J).

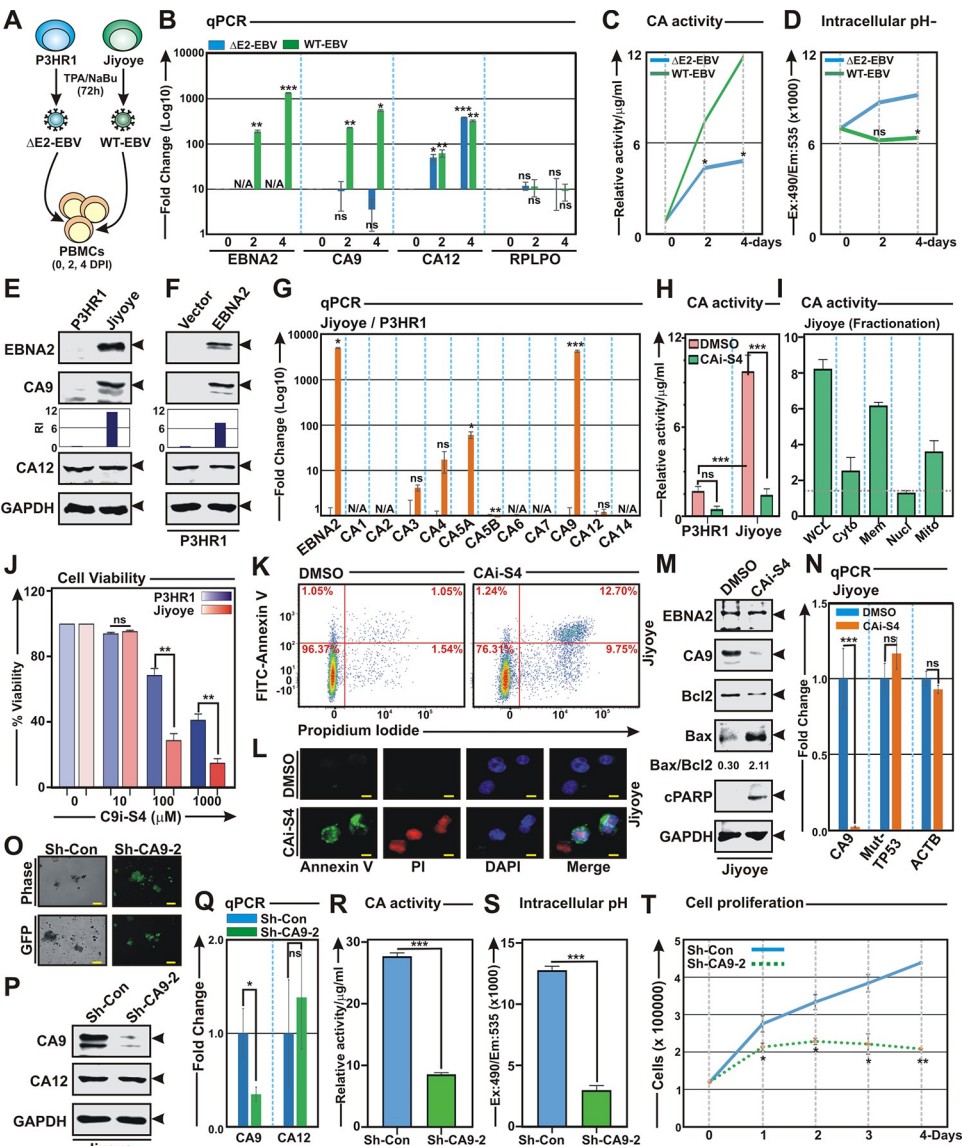

**Fig 8. EBNA2 mediated CA9 transcriptional activation is essential for B-cell survival.** (A-D) PBMCs infected (MOI: ~10) with EBNA2 deleted and wild-type EBV generated from P3HR1 and Jiyoye cells, respectively for the indicated time points were subjected to (B) qPCR, (C) carbonic anhydrase activity and (D) intracellular pH analyses. (E, G and H) Comparative analyses of P3HR1 and Jiyoye cells by (E) western blot, (G) qPCR and (H) carbonic anhydrase activity analyses with or without the treatment of 0.1 mM compound S4, respectively. (F) ~20 million P3HR1 cells were transiently transfected with empty vector or pSG5-EBAN2 construct by electroporation. 48 h post-transfection cells were harvested and subjected to western blot analyses with the indicated antibodies. (I) ~20 million Jiyoye were fractionated into nuclear, membrane, mitochondrial and cytoplasmic cell fractions using centrifugation methods, followed by carbonic anhydrase activity assay. Bar diagrams are the average of two independent experiments. (J) ~5 x $10^4$ P3HR1 or Jiyoye cells either left untreated (DMSO control) or treated with increasing concentrations (10–1000 μM) of CA9/CA12 specific inhibitors S4 for 24 h and cell viability was measured by Trypan blue exclusion method using an automated cell counter. (K-N) ~10 x $10^6$ Jiyoye cells were treated with DMSO or 0.1 mM S4 for 24 h and subjected to (K-L) cell apoptosis, (M) western blot and (N) qPCR analyses. (K-L) Representative images of (K) flow cytometry and (L) fluorescence microscopy analyses showing cell apoptosis in Jiyoye after 0.1 mM S4 treatment for 24 h. (K) FITC labelled annexin V binding and PI staining were detected using a BD cell analyzer. Results were analysed using Floreada.io, an online based tool for flow cytometry data analyses. (L) DAPI stained cell nucleus, FITC-labelled Annexin V and PI were visualized by blue (Ex = 359 nm; Em = 457 nm), green (Ex = 488 nm; Em = 350 nm) and red (Ex = 586 nm; Em = 603 nm) signals, respectively using a Ziess fluorescence microscope. Scale bars, 10 μm. (O-T) Jiyoye cells stably transduced with lentiviruses expressing either sh-control or sh-RNA (2) against CA9 gene were (O) photographed using a using a Fluorescent Cell Imager and subjected to (P) western blot, (Q) qPCR, (R)

carbonic anhydrase activity, (S) intracellular pH and (T) cell proliferation analyses. (T) For cell proliferation, ~1 x 10$^5$ Jiyoye cells stably expressing either sh-control or sh-CA9-2 were grown in 6-well plates and counted the viable cells every 24 h for the indicated time points using Trypan blue exclusion method in an automated cell counter. (C, H-I and R) Carbonic anhydrase activity and (D and S) intracellular pH determination assays were performed using kits according to the manufacturer's instructions. (E-F, M and P) Western blot analyses were performed with the indicated antibodies, where GAPDH represents as loading control. (B, G, N and Q) For qPCR analyses, the relative changes in transcripts of the selected genes using the 2$^{-\Delta\Delta Ct}$ method are represented as bar diagrams in comparison to control samples using B2M as housekeeping gene. Error bars represent standard deviations of duplicate assays of two independent experiments. *, **, *** = p-value < 0.01, 0.005 and 0.001 respectively. ns: non-significant; N/A: not amplified.

Similarly, S4 treatment resulted in more apoptotic cell death (~ 2 fold) in Jiyoye as compared to P3HR1 cells (Figs 8K and 8L and S9). Compound S4 induced cell apoptosis in Jiyoye cells was further confirmed by increased Bax/Bcl2 ratio and PARP cleavage in western blot analyses (Fig 8M). As similar to EBV transformed B-lymphocytes, compound S4 treatment resulted in significant decrease in CA9 expression in Jiyoye cells, via transcriptional repression (Fig 8M and 8N).

In order to determine whether EBNA2 mediated elevated CA9 expression is required for B-cell growth, CA9 expression was knocked-down using two specific sh-RNA sequences cloned in pGIPZ vector. The efficiency of CA9 knockdown by these two sh-RNAs in stably transduced Jiyoye cells was confirmed by western blot and qPCR analyses (Figs 8O–8Q and S10A–S10C). CA9 knock-down resulted in significant reduction of both CA activity and intracellular pH in Jiyoye cells (Figs 8R and 8S and S10D and S10E). To understand whether CA9 knock-down mediated reduction of CA activity and intracellular pH has an effect on cell growth, we conducted cell proliferation assay for 4 days (Figs 8T and S10F). As compared to the control line, CA9 knock-down Jiyoye cells showed significant retardation of cell proliferation (Figs 8T and S10F). Overall, the results indicated that EBNA2 mediated CA9 transcriptional activation is important for EBV induced nascent B-cell activation, B-cell proliferation and survival.

## EBV lytic cycle reactivation leads to CA9 downregulation

While latent EBV infection in quiescent B-lymphocytes leads to B-cell activation, which ultimately develops B-cell lymphomas [1,8], EBV reactivation into lytic replication is essential to maintain the production of progeny virus and viral transmission from one host to another [55]. By partially characterized mechanisms B-cell differentiation into plasma cells provokes EBV lytic cycle reactivation [56]. Elevated CA9 expression during EBV latent infection, prompted us to further investigate CA9 expression pattern during lytic cycle reactivation. Among various small molecules identified as stimulators of EBV lytic cycle replication [12], protein kinase C (PKC) activators, such as phorbol esters, and histone deacetylase (HDAC) inhibitors, such as sodium butyrate, are considered as the main classes of chemical inducers for lytic cycle induction in latently infected cells. Combination of 12-O-tetradecanoylphorbol-13-acetate (TPA) and sodium butyrate was utilized to induce EBV lytic cycle reactivation in two LCLs–LCL#1 and LCL#89. RNA-Seq analyses of 0, 48 and 72 h post-treatment LCLs revealed transcriptional repression of only CA9 among all fifteen CA isoforms (Fig 9A–9C). Upon reactivation, like any other herpesvirus, EBV undergoes three consecutive lytic stages— immediate early, early, and late stages [57]. The immediate early factor BZLF1, the master regulator of EBV lytic cycle, triggers expression of ~30 early lytic genes through transcriptional activation, important for lytic DNA replication [12,58]. qPCR and western blot analyses further confirmed CA9 repression and BZLF1 activation in response to chemically induced EBV lytic cycle replication in both LCLs in a dose dependent fashion (Fig 9D and Fig 9E, respectively). Downregulation of CA9 expression also directly correlated with significant decrease in CA activity during viral lytic cycle reactivation in a dose dependent manner (Fig 9F). However,

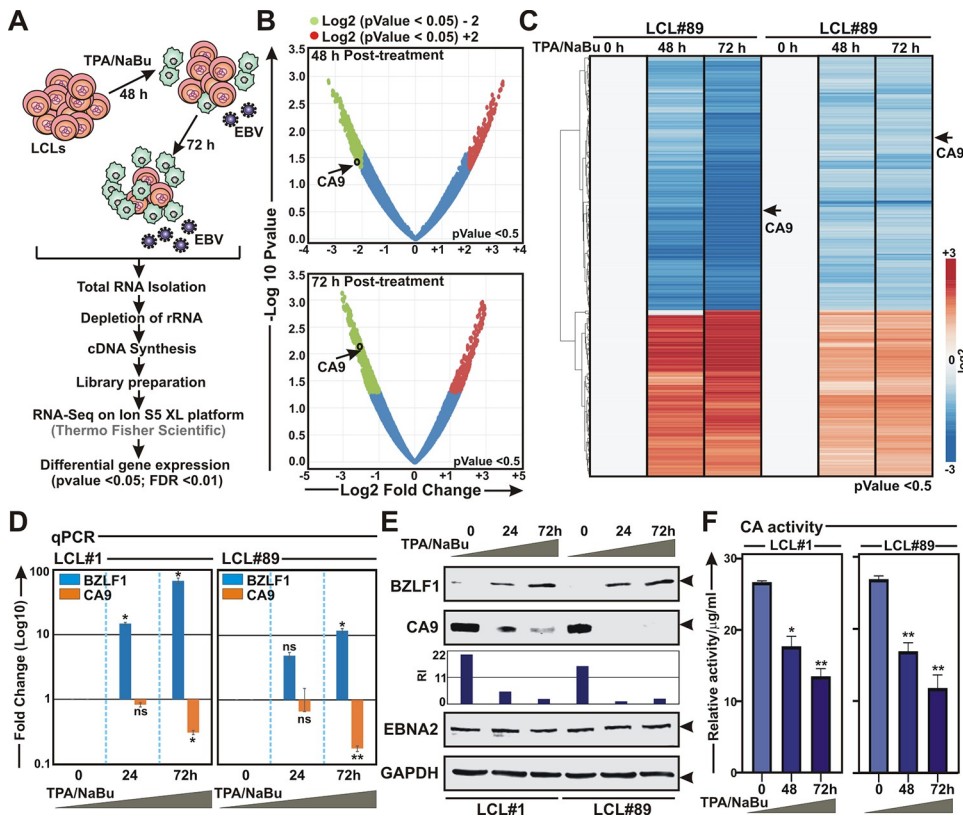

**Fig 9. RNA-Seq analyses of EBV lytic cycle reactivation from LCLs reveal CA9 downregulation.** (A-C) Whole transcriptome analysis of LCLs reactivated to lytic cycle replication through chemical induction with 3 mM sodium butyrate (NaBu) and 20 ng/ml 12-O-tetradecanoylphorbol-13-acetate (TPA) for the indicated time points using Ion S5 XL platform as described in the "Materials and Methods" section. (B-C) Volcano plot and heat map analyses of differentially expressed gene sets in 48 and 72 h post-induction. Differential gene expressions were performed based on p-value as < = 0.05 and log2 Fold Change as 3 and above (upregulated, red) and -3 and below (downregulated, blue). (D-F) LCLs reactivated to lytic cycle replication in a similar experimental set up were subjected to (D) western blot, (E) qPCR and (F) carbonic anhydrase activity analyses. (D) Western blot analyses were performed with indicated antibodies, where GAPDH was used as loading control. (E) For qPCR analyses, the relative changes in transcripts of the selected genes using the $2^{-\Delta\Delta Ct}$ method are represented as bar diagrams in comparison to control samples using B2M as housekeeping gene. (F) Carbonic anhydrase activity was performed using kit according to the manufacturer's instructions. Error bars represent standard deviations of triplicate assays of two independent experiments. *, **, *** = p-value < 0.01, 0.005 and 0.001 respectively. ns: non-significant.

these results raised the possibility that CA9 expression could be affected either by direct involvement of some specific EBV lytic proteins or simply by the compound effect of these chemical inducers. We and others previously demonstrated that global proteasomal inhibition also can initiate EBV lytic cycle replication [11,59]. Treatment with bortezomib, an FDA approved proteasome inhibitor [60], similarly demonstrated that upon lytic cycle reactivation by proteasomal inhibition prompted downregulation of both CA9 expression and its activity in LCLs (S11A and S11B Fig). Moreover, to nullify the direct effect, if any, of these chemical inducers on CA9 repression, HEK293 cells were incubated with either the combination of TPA and sodium butyrate or bortezomib and subjected for western blot analyses (S11C and S11D Fig, respectively). The results determined that neither of these chemical inducers had any direct role in alleviating CA9 expression (S11C and S11D Fig). We hypothesized that in contrast to B-cell transformation, suppression of CA9 expression and its activity might play an essential role in supporting the viral lytic cycle reactivation.

## EBV lytic cycle transactivator BZLF1 specifically downregulates CA9 expression

Accumulating evidence suggests that like many other herpesviruses, EBV lytic cycle reactivation also promotes a global host transcriptional shut-off primarily through an early protein BGLF5 nuclease activity [61,62]. Additionally, EBV during its lytic cycle replication also employs several secondary mechanisms for transcriptional and post-transcriptional regulations of host transcriptome [61]. For example, BZLF1, in addition to regulating viral gene transcriptions during lytic cycle reactivation [63], can also transcriptionally regulate expressions of multiple cell-cycle genes [64], possibly through restructuring host chromatin [30]. Next, we wanted to check whether BZLF1 induction during lytic cycle replication is directly responsible for CA9 transcriptional repression. To this end, HEK293 cells were transiently transfected with increasing concentrations of flag-tagged BZLF1 expression vector and subjected to qPCR and western blot analyses after 36 h transfection (Fig 10A and Fig 10B, respectively). The results demonstrated that BZLF1 expression was inversely correlated with CA9 expression in a dose dependent fashion at both transcript and protein levels (Fig 10A and Fig 10B, respectively). BZLF1 mediated depletion of CA9 expression in transiently transfected HEK293 cells was further corroborated with decreasing CA activity in a dose dependent manner (Fig 10C). Previously it has been shown that BZLF1 recruits Elongin B/C-Cul2/5-SOCS-box protein ubiquitin ligase complex to facilitate p53 degradation, which appears to be essential for efficient viral propagation [65]. To determine whether BZLF1 also regulates CA9 depletion in a similar fashion, HEK293 cells transiently transfected with BZLF1 expression vector were either left untreated (DMSO control) or treated with proteasomal inhibitor MG132 for 6 h (Fig 10D). Western blot analyses demonstrated that endogenous CA9 levels were noticeably depleted in the presence of BZLF1 expression irrespective of proteasomal inhibition (Fig 10D), suggesting that CA9 expression might be regulated by BZLF1 mediated transcriptional repression. To further confirm this, HEK293 cells were transiently transfected with myc-tagged CA9 with or without flag-tagged BZLF1 expression vectors (S12 Fig). In contrast to endogenous CA9, no change in expression was observed for exogenously expressed myc-CA9 (compare Figs 10B and 10D and S12), indicating BZLF1 mediated possible transcriptional regulation of CA9 expression.

ChIP-Seq reanalyses (E-MTAB-7788) of Raji cells stably expressing BZLF1 under doxycycline responsive promoter [66] revealed several distinct BZLF1 ChIP-Seq signals in the CA9 promoter region (Fig 10E). Model-based analysis of ChIP-Seq (MACS) tool for peak-calling with a p-value cut off set at p<0.001 up to ~8 Kb upstream of the TSS (transcription start site) identified two significant BZLF1 peaks at -7033 to -6430 (site 1) and -4595 to -3995 (site 2) on the CA9 promoter region (Fig 10E). BZLF1 belongs to cellular AP-1 family of transcription factors that binds to two different classes of BZLF1-responsive elements on DNA—the canonical AP-1 binding site and methylated CpG containing motifs [67,68]. Next, the two identified ChIP-Seq peaks in CA9 promoter were assessed for BZLF1 binding motifs using JASPAR tool and subsequently revealed four and three AP-1 motifs in the site 1 and site 2 BZLF1 binding regions, respectively (Fig 10E). Two ChIP-qPCR primer sets flanking the site 1 and site 2 were designed and subjected for ChIP-qPCR analyses using Jiyoye cells with or without EBV lytic cycle reactivation for 72 h using TPA/sodium butyrate combination (Fig 10F). As compared to the control, there was a significant enrichment of BZLF1 binding at both sites upon lytic cycle reactivation in Jiyoye cells (Fig 10F). As similar to LCLs, lytic cycle reactivation in Jiyoye cells for 72 h also resulted in significant downregulation of CA9 expression at both protein and transcript levels (Fig 10G and Fig 10H, respectively). Downregulation of CA9 expression was further corroborated with significant decrease in CA activity upon lytic cycle reactivation (Fig 10I).

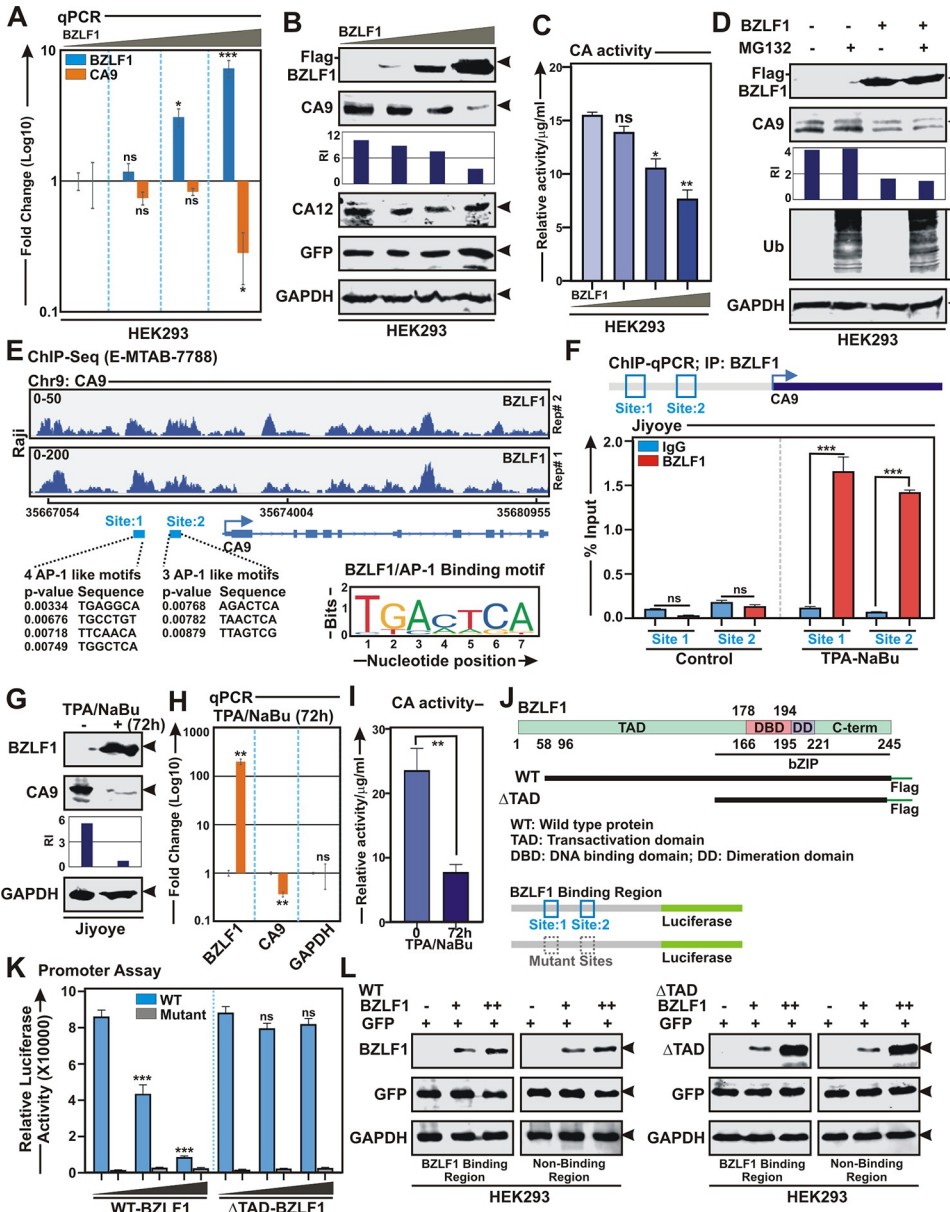

**Fig 10. EBV lytic cycle master regulator BZLF1 transcriptionally represses CA9 expression.** (A-C) ~10 x 10⁶ HEK293 cells were transiently transfected with either empty vector (pA3F) or increasing concentrations of pA3F-BZLF1 expressing flag-tagged BZLF1. 36 h post-transfection cells were harvested and subjected to (A) qPCR, (B) western blot and (C) carbonic anhydrase activity analyses. (D) HEK293 cells transiently transfected with empty vector or BZLF1 expression plasmid, were either left untreated (DMSO control) or treated with 20 μM MG132 for additional 4 h and subjected to western blot analyses. (E) Reanalyses of ChIP-Seq data (E-MTAB-7788) showing enrichment of BZLF1 on CA9 promoter region. Bottom panel indicates the MACS identified peaks (site 1 and site 2) for BZLF1 binding on CA9 promoter. Schema showing BZLF1 homologue AP-1 binding motifs on the MACS identified peaks of CA9 promoter region. (F) ChIP-qPCR data showing recruitment of BZLF1 on the CA9 promoter upon EBV lytic cycle reactivation using with 3 mM sodium butyrate (NaBu) and 20 ng/ml 12-O-tetradecanoylphorbol-13-acetate (TPA) for 72 h in Jiyoye cells. (G-I) Jiyoye cells reactivated to lytic cycle replication as similar to (F) were subjected to (G) western blot, (H) qPCR and (I) carbonic anhydrase activity analyses. (J) Schema illustrating known structural domains of BZLF1 and CA9 promoter region used for cloning in pA3F expression vector and pGL3 luciferase based reporter vector, respectively. (K) Luciferase reporter activity of the wild-type (blue) and mutant (grey) CA9 promoters in transiently transfected HEK293 cells. (L) A fraction of the total protein from (K) were evaluated by western blot analyses. (A and H) For qPCR analyses, the relative changes in transcripts of the selected genes using the $2^{-\Delta\Delta Ct}$ method are represented as bar diagrams in comparison to control samples using B2M as housekeeping gene. Error bars represent standard deviations of triplicate assays of two independent experiments. (B, D, G and L) For western blot analyses, cells were harvested, washed with 1 x PBS, lysed in RIPA and fractionated using appropriate SDS-PAGE. For

transient transfection studies, cells were additionally transfected with GFP expression vector to monitor the transfection efficiency. Western blots were performed with the indicated antibodies. GAPDH blot was performed as loading control. The relative intensities (RI) of protein bands shown as bar diagrams were quantified using the software provided by Odyssey CLx Imaging System. Representative gel pictures are shown of at least two independent experiments. (C and I) Carbonic anhydrase activity was performed using kit according to the manufacturer's instructions. Error bars represent standard deviations of triplicate assays of two independent experiments. *, **, *** = p-value < 0.01, 0.005 and 0.001 respectively. ns: non-significant.

Studies suggest that chromatin accessibility by BZLF1 strictly depends on its transactivation domain (TAD), while basic leucine-zipper (bZIP) domain alone, that is, BZLF1 lacking its TAD demonstrates similar DNA binding ability [69]. In order to directly validate BZLF1 mediated CA9 transcriptional repression, we generated ΔTAD and ΔbZIP BZLF1 constructs expressing flag-tagged BZLF1 proteins lacking TAD and bZIP regions, respectively (S13A and S13B Fig). ChIP-qPCR experiments using HEK293 cells transiently transfected either vector control or flag-tagged BZLF1 proteins further confirmed that only wild-type and ΔTAD BZLF1 proteins were significantly enriched at both site 1 and site 2 in the CA9 promoter region (S13C Fig). However, no promoter enrichment was observed for ΔbZIP BZLF1 protein (S13C Fig), indicating the importance of bZIP domain for its DNA-binding activity. Next, CA9 promoter region comprising of both site 1 and site 2 or mutant sites were inserted upstream of the luciferase gene in pGL3-basic vector and subsequently utilized for luciferase based promoter assay in the absence and presence of either wild-type or ΔTAD BZLF1 proteins (Fig 10J–10L). The results demonstrated that while wild-type BZLF1 transcriptionally repressed CA9 promoter activity in a dose dependent manner, ΔTAD BZLF1 did not show any transcriptional repression (Fig 10K).

In contrast to EBV latency programs, viral lytic cycle replication is associated with significant cell cycle arrest [70]. EBV lytic cycle induction in both epithelial and B-cell background by various agents showed a significant G1 bias [70]. Moreover, BZLF1 expression was shown to induce cell-cycle arrest at both G0/G1 and G2/M in different cell lines [71,72]. BZLF1 mediated cell cycle arrest further prompted us to examine the effect of cell-cycle arrest on CA9 expression. To this end, we first checked the cell cycle profile of EBNA2 and CA9 positive Jiyoye cells with or without the treatment of TPA/sodium butyrate for 24 h (S14A Fig). In accordance to the previously published results [71], our data also demonstrated that EBV lytic cycle induction promotes a G0/G1 cell cycle arrest (S14A Fig). In order to determine the effect of cell cycle arrest on CA9 expression, Jiyoye cells were either left untreated (DMSO control) or treated with 100 μM mimosine or nocodazole for 24 h (S14B–S14E Fig). Mimosine, a non-protein amino acid, arrests cell cycle progression at the G1-S phase, while nocodazole is a mitotic blocker that impedes microtubule polymerization. Treatment with both mimosine and nocodazole in Jiyoye cells demonstrated characteristic cell cycle blocks at G1 and G2 phases (S14B Fig and S14D Fig, respectively). In addition to flow cytometry analyses, western blot data also corroborated mimosine mediated G1 phase block by accumulation of cyclin A2 protein and nocodazole mediated mitotic arrest by increased phosphor-histone 3 as compared to the untreated sample sets (S14C Fig and S14E Fig, respectively). However, neither mimosine nor nocodazole mediated cell cycle arrests had any repressive effect on CA9 expressions (S14C and S14E Fig). The results therefore suggest that EBV lytic cycle induction mediated cell cycle arrest does not have any indirect role on CA9 repression.

Overall, the results are consistent with a model in which during EBV latent infection of nascent B-lymphocytes EBNA2 transcriptionally activates CA9 expression, while intermittent transition of latent phase to lytic replication cycle by BZLF1 transcriptionally represses CA9 expression and its activity (Fig 11).

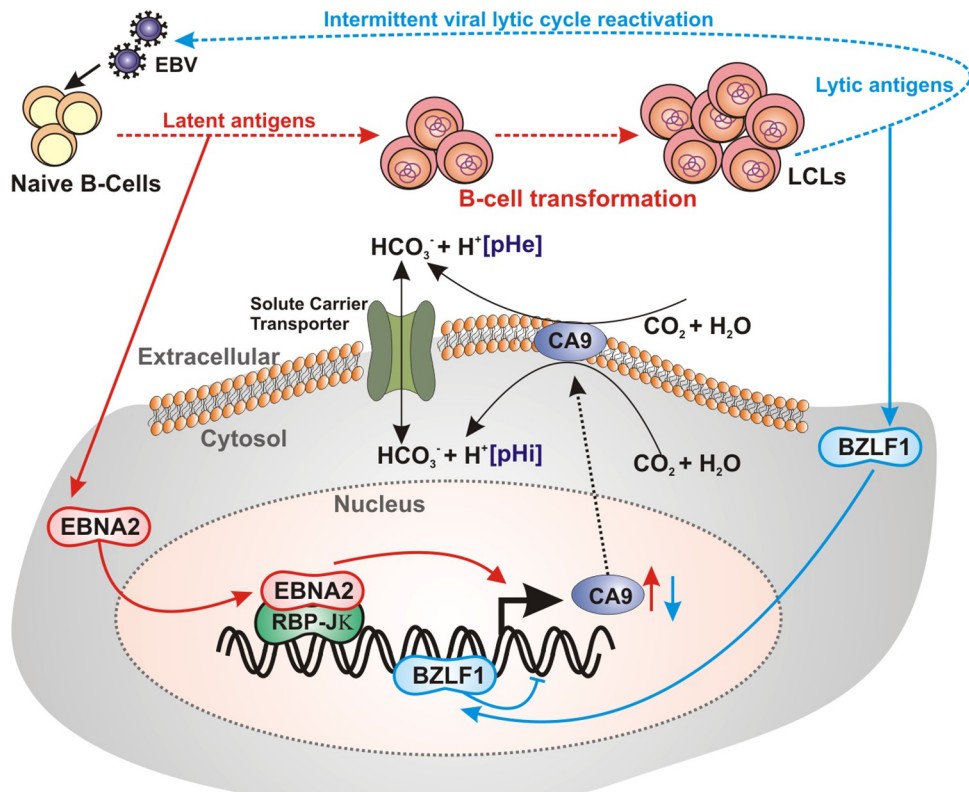

**Fig 11. Schematic representation of EBV mediated dynamic CA9 expression and pH regulation during B-cell transformation and lytic cycle reactivation.** The current model depicts while during EBV latent infection of naïve B-lymphocytes EBNA2 transcriptionally activates CA9 expression, lytic cycle reactivation by BZLF1 transcriptionally represses CA9 expression. Dynamic CA9 expression alters intracellular pH that affects viral pathogenesis and subsequent EBV-induced B-cell lymphomagenesis.

## Discussion

Metabolic reprogramming and acid-base regulation are one of the major hallmarks of cancer development [13,15]. The acid-base balance in cancer cells is controlled by the concerted mechanisms among a number of membrane transporters and carbonic anhydrase isoforms (CAs), which eventually generate an alkaline intracellular and an acidic extracellular environment [16]. The acidic shift in the extracellular tumour microenvironment fosters cancer cell proliferation by outcompeting their adjacent host cells along with increasing metastatic potential and chemotherapeutic resistance [73,74]. CA isoforms catalyse the reaction $CO_2 + H_2O \rightleftarrows HCO_3^- + H^+$ and maintain the acid-base homeostasis [16]. Out of the twelve human CAs, two membrane associated CA isoforms–CA9 and CA12 have been largely implicated in cancer development, particularly in case of solid cancers [18–24,32–34,75]. In contrast, expression profile and the associated functions of these CA isoforms have not been extensively studied in B-cell lymphoma [76]. Being a lymphotropic virus, EBV has been shown to be associated with the development of a number of B-cell malignancies [1,3,4]. Despite the strong connection with cancer cell metabolic reprogramming and acid-base balance, to date, there are no studies demonstrating a role of any CA isoforms during EBV induced B-cell transformation and subsequent B-cell lymphoma development. In this study, we demonstrated a positive correlation of elevated CA9 expression with latent EBV infection in nascent B-lymphocytes. Our results together with reanalyses of several RNA-Seq data available in the public domain further

validated that CA9 was significantly associated with EBV induced B-cell transformation, particularly during early phase of viral infection event (2–4 DPI). Studies indicated that over this period EBV elicits a hyperproliferation stage, where an elevated levels of viral oncoprotein EBNA2 through transcriptional activation of several viral (viz. LMP1) and cellular (viz. cMyc) genes supports B-cell metabolic remodelling and sustained proliferation [6,8,30,77]. In agreement to this, our study also demonstrated that EBNA2 is responsible for CA9 transcriptional activation. Genetic studies demonstrated that EBNA2 along with EBNALP, EBNA3A, EBNA3C, and LMP1 are essential for efficient B-cell transformation and LCLs growth maintenance [52,78–82]. EBNA2 and its co-transcriptional activator EBNALP are the first viral genes expressed after naïve B-cell infection [81,83]. Interestingly, although these viral oncoproteins exert their functions through transcriptional regulation, none of them are able bind DNA directly. EBNA2 typically interacts with DNA through sequence-specific transcription factor RBP-Jκ, a key mediator of Notch signalling pathway, and thereby activating cell/viral gene transcription [84,85]. Genome-wide ChIP-Seq analyses demonstrated that EBNA2 often co-localizes with RBP-Jκ and EBF1 at many promoter, enhancer as well as super-enhancer regions across the genomes in EBV transformed B-lymphocytes [85,86]. As similar to RBP-Jκ, EBF1 is also a sequence specific DNA binding protein, which cooperatively act to maintain B-cell specific transcription programs [84,87]. Reanalyses of ChIP-Seq data available in the various public domains revealed co-occupancy of EBNA2 with B-cell master regulatory factors—RBP-Jκ and EBF1 along with several other B-cell specific transcription factors including MEF2B, IKZF1, EP300 and SPI1 onto the distal CA9 promoter/enhancer region. In support of this, increased MEF2B expression was shown to be associated with several B-cell malignancies, including diffuse large B-cell lymphoma (DLBCL), a frequently EBV infected B-cell neoplasm [88]. In addition, EBNA1 transcriptionally activates EBF1 and MEF2B expressions, which play an important role for enhanced survival of EBV latently infected cells [85]. It has been shown that EBF1 along with IKZF1 and SPI1 transcription factors help to gain mature B-cell identity and control gene expression [89]. IKZF1, a zinc finger DNA-binding transcription factor, is responsible for normal lymphopoiesis and functions as a tumor suppressor protein in acute lymphoblastic leukemia [90]. Moreover, IKZF1 was shown to regulate EBV latent-lytic switch [91]. The Ets family transcription factor SPI1, is a major regulator of haematopoiesis, controlling the expression of hundreds of genes required for B-cell development [92]. Similarly, SPI1 was shown to play a major role in EBV associated B-cell lymphomagenesis particularly through regulating EBNA2 specific super-enhancer sites co-occupied with EBF1 and RBP-Jκ [49]. Both EBNA2 and its co-transcriptional activator EBNALP were shown to interact with one of the vital histone acetyltransferase enzymes EP300 [83,93]. Either EP300 or EBNALP alone is able to transcriptionally activate EBNA2 [83]. Genome-wide analyses revealed that EBNALP sites co-occupied by EP300 had significantly elevated ChIP-Seq signals for SPI1 and EBF1 among others [83]. Our analyses showed that EBNA2 was the only viral oncoprotein bound to CA9 promoter/enhancer region. Given the cooperative functions of the co-occupied B-cell specific transcription factors, it would be interesting to dissect the importance of these transcription factors discretely or in combination in regulating CA9 transcription during EBV induced B-cell transformation and subsequent B-cell lymphoma development.

Given their elevated expression in multiple cancers and prominent roles in extracellular acidification of the tumour microenvironment resulting in increasing metastatic potential and decreasing the efficacy of chemotherapy, CA9/CA12 specific inhibitors are currently emerging as potential anti-cancer drugs particularly for the treatment of solid cancers [37,38]. However, as compared to CA12, CA9 received more attention for targeted therapy due to its elevated expression in various aggressive tumours with poor clinical outcomes as well as its strong connection with hypoxic tumour microenvironment [18–24,26,32,75]. Through utilizing two

CA9/CA12 specific inhibitors–SLC-0111 and S4 along with one pan CA inhibitor acetazolamide, our data indicated that these membrane associated CA isoforms, particularly CA9, are important for EBV induced B-cell transformation and LCLs outgrowth. Most of the CA9/CA12 specific inhibitors developed over the last two decades, contain sulfamoyl structural motif. While SLC-0111, an ureido-substituted benzenesulfonamide, is under Phase Ib/II clinical trials as adjuvant agent for the treatment of advanced solid tumors [39], efficacy of S4 (4-[(3,5-Dimethylphenyl)Ureido]Phenyl Sulfamate) is currently under investigation in preclinical studies. The first generation sulfonamide CA inhibitor, acetazolamide, is a classic treatment option for metabolic alkalosis [41]. A number of studies suggest that these CA inhibitors promote cancer cell death using various cell line models through apoptotic induction [44–46]. In addition to inhibiting typical CA activity and apoptotic induction, our results further demonstrated that CA inhibitor S4 specifically blocks CA9 expression via transcriptional repression. So far, there have been no reports demonstrating this attribute of CA inhibitors. However, currently the underlying mechanism that governs CA inhibitor mediated CA9 transcriptional repression is not clear. It has earlier been demonstrated that DNA damage induced p53 facilitates Hif1α degradation and thereby downregulates CA9 expression in response to hypoxia [94]. Our results also showed that treatment with CA inhibitor increased p53 induction in LCLs, whereas no such expression was observed in case of Burkitt's lymphoma cell line Jiyoye expressing mutant form of p53. It would be interesting to study whether CA inhibitor mediated p53 induction is responsible for CA9 transcriptional repression in EBV positive B-cells expressing wild-type p53. Nevertheless, further animal studies with these CA inhibitors would further strengthen our findings. Altogether, our results suggest that CA inhibitors may have encouraging therapeutic effects against EBV associated B-cell lymphomas without having any potential cytotoxicity. The essential role of CA9 expression in EBV positive B-cells was also confirmed by knocking down CA9 expression through utilizing two specific Sh-RNAs.

As similar to all herpesviruses, EBV also elicits two alternative modes of infection—latent and lytic. While latent EBV infection of naïve B-lymphocytes causes B-cell activation, leading to B-cell blasts [1,2], intermittent lytic cycle reactivation is important for production of viral progeny as well as transmission from one host to another [55]. Accumulating evidence suggests the involvement of viral lytic cycle replication in EBV associated cancers, particularly in case of nasopharyngeal carcinoma [95]. In addition, by incompletely characterized mechanisms differentiation of B-cells into plasma cells was shown to promote EBV lytic cycle replication [56]. Given its importance in EBV pathogenesis, inhibition of EBV lytic replication may therefore serve as a treatment strategy for EBV-associated malignancies [12,96]. The strategy involves two classes of drugs–one that induces EBV lytic cycle from latency and the other that subsequently blocks lytic replication. Therefore, the success of this method relies heavily on the effectiveness of chemical inducers in reactivating EBV lytic cycle replication. To this end, a substantial effort has been made to generate various small molecules as stimulators of EBV lytic cycle replication [12]. Our results demonstrated that in contrast to EBV latent infection, lytic cycle reactivation by a combination of HDAC inhibitor sodium butyrate and protein kinase C inducer phorbol ester caused a significant decrease in CA9 expression. During reactivation into lytic cycle replication, gamma-herpesviruses, including EBV, induce a global host mRNA shut-off mechanisms [61]. This is primarily accomplished by EBV encoded early protein BGLF5 mediated degradation of host transcripts along with nuclear accumulation of cytoplasmic poly(A)-binding protein [62,97]. In addition to this canonical pathway, EBV also exerts non-canonical mechanism for blocking host gene transcriptions. For example, the immediate early lytic protein BZLF1, besides initiating viral lytic cycle replication through transcriptional activation of ~30 early lytic genes including BGLF5, can also affect host transcriptome through global restructuring of host chromatin [30]. Mechanistically, our data

showed that the BZLF1 transcriptionally repressed CA9 expression through direct binding onto its promoter region. However, at this point, we cannot rule out the involvement of BGLF5 mediated post-transcriptional control of CA9 expression. In sum, the current model portrays a differential CA9 expression during EBV pathogenesis. While during EBV latent infection elevated CA9 functions appears to be necessary for efficient B-cell transformation and survival of transformed B-lymphocytes, intermittent switch to lytic replication cycle blocks its activity. It is not yet clear, whether CA9 expression may control the EBV latent to lytic cycle transition. Future studies should focus in identifying the role of CA9 in regulating EBV latent to lytic switch and potential use of CA inhibitors for lytic induction therapy against multiple EBV associated B-cell lymphomas.

## Materials and methods

### Cell lines and cell culture

Cell lines were obtained from ATCC or received as a gift. The cell lines were further authenticated by short tandem repeat (STR)-profiling (NCCS cell repository, India). Regular testing for mycoplasma contamination was performed to confirm contamination free culture using a PCR-detection kit (Sigma-Aldrich/Merck). The details of cell lines are provided in Table 1. HEK293 and Lenti-X 293T cells were maintained in Dulbecco's modified Eagle's medium (DMEM) (Gibco/Invitrogen) supplemented with 10% fetal bovine serum (FBS) (Gibco/Invitrogen), 1% Penicillin-Streptomycin Solution (Sigma-Aldrich/Merck). HEK293T cells harboring GFP-EBV BACmid cells were maintained complete DMEM supplemented with 1 mg/ml puromycin (Sigma-Aldrich/Merck). LCLs (LCL#1 and LCL#89), P3HR1 and Jiyoye were maintained in RPMI 1640 (Gibco/Invitrogen) supplemented with 10% FBS and 1% Penicillin-Streptomycin Solution. Unless otherwise stated all above-mentioned cells were cultured at 37°C in a humidified environment supplemented with 5% $CO_2$.

### EBV stock preparation and infection of PBMCs

In order to induce EBV lytic cycle replication, HEK293T cells harboring GFP-tagged EBV-BACmid, EBV positive Burkitt's lymphoma cell lines–P3HR1 and Jiyoye were treated with 20 ng/ml 12-O-Tetradecanoylphorbol-13-acetate (TPA) and 3 mM Sodium butyrate for 5 days. Viral particle was concentrated (100 x) by ultracentrifugation at 27,000 rpm for 2 hours at 4°C and re-suspended in 0.5 ml RPMI 1640 without any supplementation. Virus was immediately used for infection or stored at -80°C. ~10 million PBMCs were in complete RPMI 1640 (0.5 million cells/ml) were incubated with GFP-EBV stock (MOI ~ 10). 24 h post-infection, cells were spun down, re-suspended in fresh media and continued growing at 37°C. Cells were harvested at the indicated time points for qPCR and western blot analyses. For B-cell transformation experiments, PBMCs isolated from two donors were infected with GFP-EBV for 28 days in 6-well plates and subsequent B-cells blasts were photographed using using a ZOE Fluorescent Cell Imager (BIO-RAD). Using similar strategy, LCLs and Jiyoye cells were reactivated to EBV lytic cycle replication by TPA and sodium butyrate treatment for 72 h and subjected to various analyses as described.

### RNA sequencing and gene ontology enrichment analysis

RNA sequencing (RNA-Seq) for PBMCs infected with GFP-EBV was performed on an Illumina NextSeq 500 platform. 0, 2, and 4-days post-infected cells were collected, total RNA was extracted using PureZOL RNA isolation reagent (BIO-RAD). A total of 1 μg of RNA that passed quality control (QC) was utilized for library preparation employing the Illumina

**Table 1. Key resources table.**

| Reagent type (species) or resource | Designation | Source | Identifiers | Additional information |
|---|---|---|---|---|
| Primary Cells (*Homo sapiens*) | PBMCs | HiMedia Laboratories Pvt.Ltd | Cat No.: CL003 | Peripheral blood mononuclear cells from multiple donors |
| Cell line (*Homo sapiens*) | HEK293T-GFP-EBV Bacmid | Gift from Dr. Erle S Robertson (University of Pennsylvania) | | HEK293T cells harbouring GFP-tagged EBV B95.8 strain |
| Cell line (*Homo sapiens*) | LCL#1 | Gift from Dr. Erle S Robertson (University of Pennsylvania) | | EBV *in vitro* transformed B-lymphocytes -lymphoblastoid cell line |
| Cell line (*Homo sapiens*) | LCL#89 | Generated from lab | | EBV *in vitro* transformed B-lymphocytes -lymphoblastoid cell line |
| Cell line (*Homo sapiens*) | P3HR1 | American Type Culture Collection (ATCC) | Cat No.: HTB-62 | Obtained from NCCS Cell repository, India. EBV positive EBNA2 deleted Burkitt's lymphoma cell line |
| Cell line (*Homo sapiens*) | Jiyoye | American Type Culture Collection (ATCC) | Cat No.: CCL-87 | Obtained from NCCS Cell repository, India. EBV positive EBNA2 deleted Burkitt's lymphoma cell line |
| Cell line (*Homo sapiens*) | HEK293 | American Type Culture Collection (ATCC) | Cat No.: CRL-1573 | Gift from Dr. Rupak Datta (Indian Institutes of Science Education and Research-Kolkata) |
| Cell line (*Homo sapiens*) | Lenti-X 293T | TaKaRa Bio | Cat No.: 632180 | Gift from Dr. Debanjan Mukhopadhyay (Presidency University, Kolkata) |
| Antibody | Anti-EBNA2 (PE2; Mouse monoclonal) | Abcam Inc. | Cat No.: ab90543 | WB: 1:1000 IF: 1:100 ChIP: 5 µg |
| Antibody | Anti-LMP1 (Rabbit monoclonal) | Abcam Inc. | Cat No.: ab136633 | WB: 1:1000 |
| Antibody | Anti-CA2, (Rabbit polyclonal) | Gift from Dr. William S Sly (Saint Louis University School of Medicine) | | WB: 1:250 |
| Antibody | Anti-CA9 (H-11; Mouse monoclonal) | Santa Cruz Biotechnology Inc. | Cat No.: sc-365900 | WB: 1:1000 IF: 1:100 |
| Antibody | Anti-CA12 (Rabbit polyclonal) | Gift from Dr. William S Sly (Saint Louis University School of Medicine) | | WB: 1:250 |
| Antibody | Anti-CA12 (D-2; Mouse monoclonal) | Santa Cruz Biotechnology Inc. | Cat No.: sc-365900 | WB: 1:1000 IF: 1:100 |
| Antibody | Anti-GAPDH (6C5; Mouse monoclonal) | Santa Cruz Biotechnology Inc. | Cat No.: sc-32233 | WB: 1:1000 |
| Antibody | Anti-GFP (Rabbit polyclonal) | Abcam Inc. | Cat No.: ab290 | WB: 1:10000 IF: 1:1000 |
| Antibody | Anti-EPAS1 (190b; Mouse monoclonal) | Santa Cruz Biotechnology Inc. | Cat No.: sc-13596 | WB: 1:1000 |
| Antibody | Anti-EPAS1 (BL-95-1A2; Rabbit monoclonal) | Abcam Inc. | Cat No.: ab243861 | ChIP: 5 µg |
| Antibody | Anti-EBNA3A (Sheep polyclonal) | Abcam Inc. | Cat No.: ab16126 | WB: 1:1000 |
| Antibody | Anti-EBNA3C (Sheep polyclonal) | Abcam Inc. | Cat No.: ab16128 | WB: 1:1000 |
| Antibody | Anti-Hif1a (H1alpha 67; Mouse monoclonal) | Santa Cruz Biotechnology Inc. | Cat No.: sc-53546 | WB: 1:200 |
| Antibody | Anti-Histone H3 (Rabbit polyclonal) | Abcam Inc. | Cat No.: ab1791 | WB: 1:5000 |
| Antibody | Anti-p53 (DO-1; Mouse monoclonal) | BD Pharmingen | Cat No.: 554293 | WB: 1:1000 |
| Antibody | Anti-Cleaved PARP (F21-852; Mouse monoclonal) | BD Pharmingen | Cat No.: 552596 | WB: 1:500 |

(*Continued*)

**Table 1.** (Continued)

| Reagent type (species) or resource | Designation | Source | Identifiers | Additional information |
|---|---|---|---|---|
| Antibody | Anti-RBP-Jκ (Rabbit polyclonal) | Proteintech | Cat No.: 14613-1-AP | ChIP: 5 µg |
| Antibody | Anti-Bcl-2 (Rabbit monoclonal) | Cell Signaling Technology | Cat No.: D55G8 | WB: 1:5000 |
| Antibody | Anti-Bax (Rabbit monoclonal) | Cell Signaling Technology | Cat No.: D2E11 | WB: 1:5000 |
| Antibody | Anti-BZLF1/ZEBRA (BZ1; Mouse monoclonal) | Santa Cruz Biotechnology Inc. | Cat No.: sc-53904 | WB: 1:1000 ChIP: 5 µg |
| Antibody | Anti-Flag (M2; Mouse monoclonal) | Sigma-Aldrich/Merck | Cat No.: F3165 | WB: 1:1000 |
| Antibody | Anti-Ubiquitin (P4D1; Mouse monoclonal) | Santa Cruz Biotechnology Inc. | Cat No.: sc-8017 | WB: 1:1000 |
| Antibody | Anti-Myc (9E10; Mouse monoclonal) | Abcam Inc. | Cat No.: ab32 | WB: 1:1000 |
| Antibody | Anti-Cyclin A2 (Mouse monoclonal) | Cell Signaling Technology | Cat No.: 4656S | WB: 1:2000 |
| Antibody | Anti-Phospho-Histone H3 (Rabbit polyclonal) | Cell Signaling Technology | Cat No.: 9701S | WB: 1:1000 |
| Antibody | Mouse IgG Isotype Control | Invitrogen/Thermo Fisher Scientific | Cat No.: 10400C | ChIP: 1:10000 |
| Antibody | Rabbit IgG Isotype Control | Invitrogen/Thermo Fisher Scientific | Cat No.: 02–6102 | ChIP: 1:10000 |
| Antibody | Goat anti-Mouse IgG, DyLight 680 | Invitrogen/Thermo Fisher Scientific | Cat No.: 35518 | WB: 1:10000 |
| Antibody | Goat anti-Mouse IgG, DyLight 488 | Invitrogen/Thermo Fisher Scientific | Cat No.: 35502 | WB: 1:10000 |
| Antibody | Goat anti-Rabbit IgG, DyLight 488 | Invitrogen/Thermo Fisher Scientific | Cat No.: 35552 | WB: 1:10000 |
| Antibody | Goat anti-Rabbit IgG, DyLight 680 | Invitrogen/Thermo Fisher Scientific | Cat No.: 35569 | WB: 1:10000 |
| Antibody | Rabbit anti-Sheep IgG, DyLight 680 | Invitrogen/Thermo Fisher Scientific | Cat No.: SA5-10058 | WB: 1:10000 |
| Antibody | Rabbit anti-Sheep IgG, DyLight 488 | Invitrogen/Thermo Fisher Scientific | Cat No.: SA5-10054 | WB: 1:10000 |
| Antibody | Goat anti-Rabbit IgG, Alexa Fluor 488 | Invitrogen/Thermo Fisher Scientific | Cat No.: A-11008 | IF: 1:500 |
| Antibody | Goat anti-Mouse IgG, Alexa Fluor 488 | Invitrogen/Thermo Fisher Scientific | Cat No.: A-11029 | IF: 1:500 |
| Antibody | Goat anti-Rabbit IgG, Alexa Fluor 568 | Invitrogen/Thermo Fisher Scientific | Cat No.: A-11011 | IF: 1:500 |
| Antibody | Goat anti-Mouse IgG, Alexa Fluor 568 | Invitrogen/Thermo Fisher Scientific | Cat No.: A-11004 | IF: 1:500 |
| Plasmid | pGIPZ- sh-Control | Gift from Prof. Erle S Robertson (University of Pennsylvania) | | Sh-RNA sequence containing 5′-TCTCGCTTGGGCGAGAGTAAG-3′ cloned into pGIPZ vector using *Xho*I/*Mlu*I |
| Plasmid | pGIPZ- shCA9-1 | Designed and cloned for this study. | | First Sh-RNA sequence containing 5'-GGAAGAAATCGCTGAGGAA-3' cloned into pGIPZ vector using *Xho*I/*Mlu*I |
| Plasmid | pGIPZ- shCA9-2 | Designed and cloned for this study. | | Second Sh-RNA sequence containing 5′-GCAACAATGGCCACAGTGT-3′ cloned into pGIPZ vector using *Xho*I/*Mlu*I |

(*Continued*)

**Table 1.** (Continued)

| Reagent type (species) or resource | Designation | Source | Identifiers | Additional information |
|---|---|---|---|---|
| Plasmid | pSG5-EBAN2 | Gift from Prof. Paul J Farrell (Imperial College London) | | EBNA2 cDNA cloned into pSG5 vector |
| Plasmid | pSG5-BZLF1 | Addgene | Cat No.: 72637 | Gift from Prof. S. Diane Hayward (Johns Hopkins University School of Medicine) |
| Plasmid | pA3F-BZLF1 | Designed and cloned for this study. | NCBI Ref Seq: NC_007605.1 | BZLF1 cDNA cloned into pCDNA3.1-3X-Flag vector using *Bam*HI/*Not*I |
| Plasmid | pA3F-ΔTAD-BZLF1 | Designed and cloned for this study. | NCBI Ref Seq: NC_007605.1 | BZLF1 cDNA lacking transactivation domain (TAD) cloned into pCDNA3.1-3X-Flag vector using EcoRI/NotI |
| Plasmid | pA3F-ΔbZIP-BZLF1 | Designed and cloned for this study. | NCBI Ref Seq: NC_007605.1 | BZLF1 cDNA lacking DNA binding bZIP domain cloned into pCDNA3.1-3X-Flag vector using EcoRI/NotI |
| Plasmid | pGL3-CA9p-WT | Designed and cloned for this study. | NCBI Ref Seq: NC_000009.12 | Wild-type CA9 promoter region cloned into pGL3-basic vector using *Mlu*I/*Bgl*II |
| Plasmid | pGL3-CA9p-Mut | Designed and cloned for this study. | NCBI Ref Seq: NC_000009.12 | Mutant CA9 promoter region generated by assembly PCR and cloned into pGL3-basic vector using *Mlu*I/*Bgl*II |
| Plasmid | pEGFP-C1 | Clontech Laboratories, Inc | Cat No.: 632470 | Plasmid containing GFP cDNA |
| Plasmid | pCMV3-C-FLAG-CA9 | Sino Biological Inc | Cat No.: HG10107-CF | Plasmid containing CA9 cDNA cloned into pCMV3-C-FLAG vector |
| Plasmid | pA3M-CA9 | Designed and cloned for this study. | NCBI Ref Seq: NM_001216.2 | CA9 cDNA cloned into pCDNA3.1-3X-Myc vector using EcoRI/*Not*I |
| Plasmid | psPAX2 | Addgene | Cat No.: 12260 | Gift from Didier Trono (Swiss Federal Institute of Technology in Lausanne); 2nd generation lentiviral packaging plasmid |
| Plasmid | pMDG | Addgene | Cat No.: 187440 | Gift from Simon Davis (University of Oxford); VSV-G envelope expressing plasmid |
| Chemical compound, drug | 12-O-Tetradecanoylphorbol-13-acetate (TPA) | Sigma-Aldrich/Merck | Cat No.: P1585 | |
| Chemical compound, drug | Sodium butyrate | Sigma-Aldrich/Merck | Cat No.: B5887 | |
| Chemical compound, drug | ODN 2006 | InvivoGen | Cat No.: tlrl-2006-1 | Class B CpG oligonucleotide—Human TLR9 ligand |
| Chemical compound, drug | 4',6-Diamidino-2-Phenylindole, Dihydrochloride (DAPI) | Invitrogen/Thermo Fisher Scientific | Cat No.: D1306 | Nuclear staining for immunofluorescence studies |
| Chemical compound, drug | SLC-0111/U-104 | MedChemExpress | Cat No.: HY-13513 | Carbonic anhydrase inhibitor |
| Chemical compound, drug | S4 | MedChemExpress | Cat No.: HY-110243 | Carbonic anhydrase inhibitor |
| Chemical compound, drug | Acetazolamide | MedChemExpress | Cat No.: HY-B0782 | Carbonic anhydrase inhibitor |
| Chemical compound, drug | Polybrene | Sigma-Aldrich/Merck | Cat No.: TR-1003-G | Reagent for lentivirus transduction |

(*Continued*)

**Table 1.** (Continued)

| Reagent type (species) or resource | Designation | Source | Identifiers | Additional information |
|---|---|---|---|---|
| Chemical compound, drug | Puromycin dihydrochloride | Sigma-Aldrich/Merck | Cat No.: P8833 | Antibiotic selection for stable cell line generation |
| Chemical compound, drug | MG132 | Abcam Inc. | Cat No.: ab141003 | Proteasomal inhibitor |
| Chemical compound, drug | bortezomib | Abcam Inc. | Cat No.: ab142123 | Proteasomal inhibitor |
| Chemical compound, drug | Mimosine | MedChemExpress | Cat No.: HY-N0928 | Induces cell cycle arrest at the G0/G1 phase |
| Chemical compound, drug | Nocodazole | Sigma-Aldrich/Merck | Cat No.: M1404 | Induces mitotic arrest during cell cycle |
| Chemical compound, drug | Lipofectamin 3000 | Invitrogen/Thermo Fisher Scientific | Cat No.: L3000008 | Transfection reagent |
| Chemical compound, drug | jetPRIME | Polyplus Transfection Inc. | Cat No.: 101000015 | Transfection reagent |
| Software, algorithm | DAVID v6.8 | DAVID Bioinformatics | | https://david.ncifcrf.gov/ |
| Software, algorithm | Microsoft Excel 2013 | Microsoft | | www.microsoft.com/ |
| Software, algorithm | GraphPad Prism | GraphPad Software | | www.graphpad.com |
| Software, algorithm | ZEN | ZEISS | | www.zeiss.com/ |
| Software, algorithm | Image Studio v2.0 | LI-COR Biosciences | | www.licor.com/bio/image-studio/ |
| Software, algorithm | CFX- Maestro v2.3 | BIO-RAD | | www.bio-rad.com/ |
| Software, algorithm | GTEx | GTEx Portal | | https://gtexportal.org/home/ |
| Software, algorithm | ImageJ2 | ImageJ2 | | https://imagej.nih.gov/ij/ |
| Software, algorithm | Floreada.io | Floreada.io | | https://floreada.io/analysis |
| Software, algorithm | ImageLab v5.1 | BIO-RAD | | www.bio-rad.com/ |
| Software, algorithm | Integrative Genomics Viewer (IGV) | Broad Institute | | https://igv.org/app/ |
| Software, algorithm | ChIP-Atlas | Kyoto University | | https://chip-atlas.org/ |
| Software, algorithm | JASPAR | JASPAR 2024 | | https://jaspar.genereg.net/ |
| Software, algorithm | MACS | | | https://pypi.org/project/MACS/ |
| Software, algorithm | DepMap | Broad Institute | | https://depmap.org/portal/ |
| Software, algorithm | FlowJo_v10.8.1 | BD Biosciences | | https://www.flowjo.com/ |

TruSeq standard mRNA sample prep kit. After quality assessment of the prepared libraries using the Tape Station 4200 (Agilent Technologies), the QC-approved libraries were subsequently sequenced. For read quality control, FASTQC was employed, and high-quality reads were processed through Trimmomatic v0.35 to eliminate adapter sequences. The trimmed sequences were then aligned to the human reference genome (Homo sapiens.GRCh37) using BWA v0.7.12 with default parameters.

RNA-Seq for EBV lytic cycle reactivation samples were performed on a S5 XL System (Thermo Fisher Scientific Inc.). Total RNA was isolated from 0, 48 and 72 h post-reactivated LCLs (LCL#1 and LCL#89) PureZOL RNA isolation reagent and subjected for whole transcriptome enrichment through selective depletion of 99.9% of ribosomal RNA using RiboMinus Eukaryote Kit for RNA-Seq (Thermo Fisher Scientific Inc.) followed by library generation using Ion Total RNA-Seq Kit v2 (Thermo Fisher Scientific Inc.). Reads generated from Ion S5 XL were subjected to further analysis with a minimum read length of 25bp. The filtered in reads were aligned to Human genome (Homo sapiens.GRCh37) using two-step alignment method. Reads were aligned using STAR and Bowtie software using default parameters. RNASeqAnalysis plugin (v5.2.0.5) was utilized to perform the analysis and produce gene counts for all the samples.

Differential expression analysis between two samples was conducted using DESeq2 package from R. Genes were selected based on p-value as $< = 0.05$ and $\log_2$ Fold Change as 2 and above (up-regulated) and -2 and below (down-regulated). Differentially expressed gene sets were segregated together and subsequently uploaded on DAVID v6.8 webserver for further analyses. Features observed in different databases were clustered together for functional analysis. Gene Ontology (GO) was selected from the hits table for DAVID clustering. The clusters were represented through statistical analysis providing p-value and FDR (false discovery rate) in order to select functional significant clusters.

## Real-time quantitative PCR (qPCR) and western blot analyses

Total RNA was extracted from ~10 x $10^6$ cells of each experiment using PureZOL RNA Isolation reagent following the manufacturer (BIO-RAD) protocol. Subsequently, cDNA was synthesized using iScript cDNA synthesis kit (BIO-RAD) according to the manufacturer's instructions. The quality and quantity of both RNA and cDNA were evaluated using Synergy H1 Multimode Microplate Reader (BioTek). For qPCR analysis, the iTaq Supermix (BIO-RAD) was utilized in the CFX Connect real-time PCR detection system (BIO-RAD). The qPCR thermal profile consisted of one cycle at 95˚C for 30 seconds, followed by 40 cycles of 95˚C for 5 seconds and 60˚C for 30 seconds. Additionally, a melt curve was generated by gradually increasing the temperature from -65˚C to 95˚C in 0.5˚C increments, with a duration of 5 seconds per step. Unless specified otherwise, each reaction was performed in triplicate, and the relative transcript levels were determined using the $2^{-\Delta\Delta CT}$ method, normalized to a housekeeping gene control. The sequences of primers used for real-time PCR are given in S3 Table.

For western blot analyses, ~10 x $10^6$ cells from each experiment were harvested, washed with ice cold $1 \times$ PBS (Gibco/Invitrogen), and subsequently lysed in 0.5 ml ice cold RIPA buffer (Thermo Fisher Scientific Inc.) with 1 x protease inhibitor cocktail (Cell Signaling Technology Inc.). Protein samples were estimated by Bradford reagent (BIO-RAD). Samples were boiled in 4 x Laemmli buffer (BIO-RAD), resolved by appropriate SDS-PAGE and transferred to a 0.45 μm nitrocellulose membrane (BIO-RAD). The membranes were then probed with specific primary antibodies followed by incubation with appropriate DyLight secondary antibodies and viewed on an Odyssey CLx Imaging System (LiCor Inc.). Image analysis and quantification measurements were performed using the Image Studio (LiCor Inc.). The list of primary and secondary antibodies used in western blot analyses are given in Table 1.

## Fluorescence microscopy

Following each experimental procedure, cells were collected via centrifugation and subjected to two rounds of washing with 1 x PBS. Cells were then fixed and permeabilized using a combination of 4% paraformaldehyde (Sigma-Aldrich/Merck) and 0.1% Triton X-100 (Sigma-Aldrich/Merck). After blocking with 5% BSA (BIO-RAD) in 1 X PBS at room temperature for 1 h, cells were incubated with appropriate primary antibodies at room temperature for 2 h, followed by incubation with species specific Alexa Fluor secondary antibody (Thermo Fisher Scientific Inc.) at room temperature for 1 h. Nuclei were counterstained with 4', 6',-diamidino-2-phenylindole (DAPI; BIO-RAD) at room temperature for 30 minutes. Cells were then washed three times with 1 x PBS and mounted using an antifade mounting media (Sigma-Aldrich/Merck). The images were captured using a Zeiss AXIO Observer.Z1 Inverted Fluorescence microscope (Zeiss), and subsequently analysed using ZEN software (Zeiss). The list of primary and secondary antibodies used in immunofluorescence analyses are given in Table 1.

## Sub-cellular fractionation

~10 million LCLs or Jiyoye were harvested and suspended into 500 μL fractionation buffer containing 20 mM HEPES (HiMedia Laboratories) pH 7.4, 10 mM KCl, 2 mM MgCl$_2$, 1 mM EDTA (MedChemExpress), 1 mM EGTA (MedChemExpress), 1 mM DTT (Abcam Inc.) supplemented with 1 x protease inhibitor cocktail. After incubation on ice for 15 minutes, the cell suspension was then passed through a 27-gauge needle for 10 times. Following incubation on ice for 20 minutes, the sample was centrifuged at 720 g for 5 minutes. The resulted pellet contained nuclear fraction while the supernatant contained cytoplasmic, membrane and mitochondrial fractions. The supernatant was then transferred to a fresh tube and kept on ice for further processing. The nuclear pellet was washed with 500 μl fractionation buffer, passed through a 25-gauge needle and centrifuged at 720 g for 10 minutes to obtain purified nuclear fraction as pellet. The pellet re-suspended in 1 x Tris buffer saline (TBS) with 0.1% SDS (BIO-RAD) was subjected to a brief sonication on ice for shearing the genomic DNA using Diagenode Bioruptor Plus sonicator (Diagenode Inc.). The supernatant was further centrifuged at 10000 x g for 5 minutes, yielding mitochondrial fraction as pellet. The mitochondrial pellet was processed similarly to the nuclear pellet. The supernatant comprising of both cytoplasmic and membrane fractions, was transferred to a fresh tube for ultracentrifugation at 100000 x g for 1 h. The resulting pellet was re-suspended in fractionation buffer and passed through a 25-gauge needle. After re-centrifugation at 100000 x g for 45 minutes, the pellet containing membrane fraction was processed as similar to nuclear and mitochondrial fractions. The remaining supernatant contained cytoplasmic fraction. All the fractions were boiled with 4 x sample buffer for western blot analyses with the indicated antibodies.

## Carbonic anhydrase (CA) activity and intracellular pH assays

CA activity was performed using a colorimetric based assay kit (Abcam Inc.) according to manufacturer's instructions. Briefly, ~5 million cells of individual experiment were harvested, washed twice with ice-cold saline solution (1 mM Tris, pH 8.0, 200 mM NaCl) by centrifugation at 3000 x g for 5 min at 4°C. Cells were lysed in 1 mM Tris, pH 8.0 buffer by incubation on ice for 10 minutes followed by -80°C for another 15 minutes. Supernatant was collected by centrifugation at 15000 g for 15 minutes. The total protein was estimated by Bradford reagent and normalised accordingly. 10 μl protein samples (2.5 μg) in each well of a 96-well plate (Corning Inc.) was diluted by 10 x CA Assay Buffer to make volume up to 95 μl. After adding 5 μl CA substrate to each well, absorbance at 405 nm was recorded for 1 h in a kinetic mode using a microplate reader (BioTek). Max-V was plotted as relative CA activity.

Intracellular pH was measured using fluorescent based assay kit (Abcam Inc.). Briefly, for each experiment, ~5 x $10^5$ cells were harvested, re-suspended in 100 μl fresh media and plated in each well of a 96-well plate. After addition of 100 μl of cell-permeable fluorescent indicator BCFL-AM dye-loading solution, 96-well plate was incubated at 37˚C for 30 minutes followed by incubation at room temperature for another 30 minutes. Fluorescence was measured in a microplate reader (BioTek) with excitation and emission at 490 and 535 nm, respectively.

## Cell viability assay

~$0.5 \times 10^5$ cells plated into each well of the six-well plates (Corning Inc.) were treated with carbonic anhydrase inhibitors–SLC-0111/U-104, S4 and acetazolamide with increasing concentrations (0–1000 μM) for the indicated time points at 37˚C in a humidified $CO_2$ chamber (Thermo Fisher Scientific Inc.) or in a hypoxic environment in the presence of 1% $O_2$ in humidified $CO_2$ chamber (New Brunswick Scientific/Eppendorf) or in the presence of 300 mM $CoCl_2$ (Sigma-Aldrich/Merck). Viable cells from each well were measured by Trypan blue (Gibco/Invitrogen, Inc.) exclusion method using an automated cell counter (BIO-RAD). Experiments were performed in triplicate and were independently repeated at least two times unless otherwise stated.

## Colony formation assay using soft-agar

0.75% agar (Sigma-Aldrich/Merck) in 1 ml complete RPMI 1640 medium was poured into each well of 6-well plates (Corning Inc.) and set aside to solidify. Next, ~$1 \times 10^5$ LCLs either left untreated (DMSO control) or treated with increasing concentrations of carbonic anhydrase inhibitors (10, 100 and 1000 μM) for 24 h, were harvested, mixed with 1 ml complete RPMI 1640 medium supplemented with 0.36% agar and poured on the top of the solidified 0.75% hard agar layer in each well. Two weeks later, colonies were stained with 0.1% crystal violet (Sigma-Aldrich/Merck) for 30 minutes and photographed (bright-field) using a ZOE Fluorescent Cell Imager (BIO-RAD). The relative colony number was measured using ImageJ2 software.

## Reanalysis of RNA-Seq data

The raw FASTQ files of RNA-Seq data (GSE125974 and DRA011328) of EBV infection of primary B-lymphocytes were downloaded from respective repositories and analyzed using Galaxy webserver (https://usegalaxy.org/). Similarly, FASTQ RNA-Seq reads of PBMC (accession ID: SRR7251667, SRR7251668, SRR7251669, SRR7251670) and GM12878 LCLs (accession ID: SRR14638511, SRR14638512, SRR306999, SRR307000, SRR307001, SRR307002, SRR307003) were downloaded from the Sequence Read Archive (SRA) (https://www.ncbi.nlm.nih.gov/sra/). The abundance of transcripts were quantified using Kallisto quantification tool (v. 0.48.0) into transcripts per million (TPM). The log converted TPM data were visualized using GrapPad Prism v8.

RNA-Seq data of EBV positive and negative cell lines were downloaded from DepMap portal (https://depmap.org/portal/). Expression data was extracted in transcripts per million (TPM) from the DepMap Expression Public 23Q4 dataset and further visualized using GrapPad Prism v8.

## Apoptosis assay

Cell apoptosis resulting from the treatment with carbonic anhydrase inhibitor–S4 was evaluated using the Annexin V-FITC Apoptosis Staining/Detection Kit (ab14085) according to the

manufacturer's protocol (Abcam Inc.). Briefly, ~5 x $10^5$ LCLs (LCL#1 and LCL#89), Jiyoye and P3HR1 cells were either left untreated (DMSO control) or treated with 0.1 mM S4 for 24 h. Post-treated cells were collected by centrifugation at 1800 rpm for 5 minutes, re-suspended in 500 μl of 1 x binding buffer, and stained with 5 μl each of FITC labelled annexin V and propidium iodide (PI; Sigma-Aldrich/Merck) for 5 minutes at room temperature in the dark. The annexin V binding and PI staining were measured using the FITC signal detector (Ex = 488 nm; Em = 350 nm) and the phycoerythrin emission signal detector (Em = 574 nm), respectively, in BD LSRFortessa (BD Biosciences). The data was analysed using Floreada.io web-based software.

A fraction of these stained cells were also subjected for fluorescence microscopy analyses. Nuclei were additionally labelled with Hoechst 3342 solution (Thermo Fisher Scientific Inc.) at room temperature for 30 minutes. Cells were washed three times with 1 x PBS and mounted using an antifade mounting media (Sigma-Aldrich/Merck). The images were captured using a Zeiss AXIO Observer.Z1 Inverted Fluorescence microscope (Zeiss), and subsequently analysed using ZEN software (Zeiss). Two independent experiments were carried out in similar settings and representative data were shown in the figures.

## Semi-quantitative reverse transcription PCR

~ 50 ng cDNAs generated from total RNA of both LCLs–LCL#1 and LCL#89 either left untreated (DMSO control) or treated with 0.1 mM carbonic anhydrase inhibitor–S4 for 24 h was subjected to semi-quantitative RT-PCR analyses using Q5 high-fidelity DNA polymerase (New England Biolabs Inc.) according to manufacturer's protocol in a T100 Thermal cycler (BIO-RAD). The similar primer sets (S3 Table) used for qPCR analyses were utilized for PCR amplification. Amplified PCR products were ran on 2% agarose gel and stained with ethidium bromide solution (New England Biolabs Inc.). Gel photographs were taken using ChemiDoc MP gel documentation system (BIO-RAD). The primers used semi-qPCR are available in S3 Table.

## Reanalyses of ChIP-Seq and ChIA-PET data

Reanalyses of ChIP-Seq data listed in S2 Table were carried out using MACS2 (model-based analyses of ChIP-Seq) algorithms for sample replicates downloaded from corresponding SRA files against the matching input file. Fastq files were extracted from SRA files and aligned against human reference genome (Homo sapiens.GRCh37) using Strand NGS with default parameters. BAM files were assessed for duplicate reads and subsequently removed. A processed non-redundant BAM file was then analysed to identify peaks using MACS2 with default parameters. For subsequent analysis, peaks with a false discovery rate (FDR) of <1% were selected. Replicates were combined by retaining only those peaks that were present in both samples. p-values were calculated based on hypergeometric distribution and corrected for multiple testing using Bonferonni correction.

RNA-Pol II ChIA-PET dataset (GSE158897) of GM12878 cell line was downloaded from Gene Expression Omnibus (https://www.ncbi.nlm.nih.gov/geo/) and visualized using WashU Epigenome Browser (https://epigenomegateway.wustl.edu/).

## ChIP-qPCR analyses

~20 million cells were cross-linked with 1% (vol/vol) formaldehyde (Sigma-Aldrich/Merck) for 10 minutes at room temperature. After quenching the cross-linking reaction by 125 mM glycine for 5 minutes at room temperature, the cells were washed with ice cold 1 × PBS and suspended in lysis buffer (50 mM Tris-HCl pH 8.1, 10 mM EDTA, 1% SDS and 1 × protease

inhibitor cocktail). To attain chromatin fragments of ~200–500 bp, cells were sonicated using Diagenode Bioruptor Plus sonicator (Diagenode Inc.) with 20 sets of 30 seconds pulses in ice. After saving a 10% of the solubilzed chromatin as input, the sheared chromatin was diluted to 10-fold in immunoprecipitation (IP) dilution buffer (16.7 mM Tris-HCl pH 8.1, 1.2 mM EDTA, 167 mM NaCl, 1.1% Triton X-100, 0.01% SDS along with 1 × protease inhibitor cocktail), followed by immunoprecipitation with 5 µg appropriate antibodies or corresponding IgG control using magnetic protein A/G beads (BIO-RAD). After several rounds of washing with low-salt wash buffer (20 mM Tris-HCl, pH 8.1, 2 mM EDTA, 150 mM NaCl, 1% Triton X-100, 1% SDS), high-salt wash buffer (20 mM Tris-HCl, pH 8.1, 2 mM EDTA, 500 mM NaCl, 1% Triton X-100, 1% SDS), LiCl wash buffer (10 mM Tris-HCl, pH 8.1, 1 mM EDTA, 0.25 M LiCl, 1% NP-40, 1% deoxycholate acid), and TE buffer (10 mM Tris-HCl, pH 8.1, 1 mM EDTA), the protein-DNA complexes were eluted using elution buffer (100 mM NaHCO3, 1% SDS) at 55˚C for 15 minutes. After reverse cross-linking by proteinase K (Thermo Fisher Scientific Inc.) treatment (40 U/ml), the ChIP-ed DNA was purified using the QIAquick PCR purification kit (QIAGEN) and subjected for qPCR analyses. The data was normalized as a percentage of the input DNA. The primers for ChIP-qPCR are available in S3 Table.

## Transient transfection in epithelial and B-cells

~10 x 10$^6$ HEK293 or ~20 x 10$^6$ P3HR1 cells were harvested, re-suspended with 450 µL Opti-MEM medium (Gibco/Invitrogen), mixed with appropriate plasmids in a 0.4-cm gap cuvette (BIO-RAD) and subjected to electroporation using a Gene Pulser II electroporator (BIO-RAD) at 210V (for HEK293) or 230V (for P3HR1) and 975µF. Cells were harvested 36 h post-transfection for western blot, qPCR and CA activity analyses. For luciferase based promoter assays, HEK293 cells were transfected with appropriate plasmids using either Lipofectamine 3000 (Thermo Fisher Scientific Inc.) or JetPrime (Polyplus Transfection Inc.) according to manufacturer's protocol. For lentivirus generation, Lenti-X 293T cells were transfected using CaPO$_4$ method. A DNA/CaPO4 mix was prepared by adding appropriate plasmid DNAs with 62 µl of 2 M CaCl$_2$ and 500 µl of 2 x HBS (280 mM NaCl, 10 mM KCl, 1.5 mM Na$_2$HPO$_4$, 12 mM Dextrose, 50 mM HEPES, pH 7.05) and adjusted the volume up to 1 ml with sterile H$_2$O. The mixture was allowed to settle for 30 minutes at room temperature and added on to the top of the plated cells followed by incubation for overnight.

## Lentivirus mediated knockdown of CA9 in B-cells

Lenti-X 293T cells were transfected with pGIPZ-sh-Cont or two different pGIPZ-sh-CA9 clones in the presence of lentivirus packaging vectors—pMDG and psPAX2 using CaPO$_4$ method. 12 h post-transfection the media was removed and fresh media was added with 3 mM Sodium butyrate (Sigma-Aldrich, USA) to induce lentivirus replication. 48 h post-incubation lentivirus containing media was collected, mixed with ~5 x 10$^5$ Jiyoye cells in the presence of 8 µg/ml polybrene (Sigma-Aldrich/Merck) and spun at 800 g for 2 h. Transduced cells were selected with 1 µg/ml Puromycin (Sigma-Aldrich, USA) in complete RPMI for 7-days. Oligo sequences for CA9 knockdown are available in S3 Table.

## Luciferase-based promoter assay

Promoter assays were performed using Dual-Glo Luciferase Assay Systems kit (Promega) according to manufacturer's protocol. Briefly, ~5 x 10$^5$ HEK293 cells grown in 12-well plates (Corning Inc.) were transiently transfected with pGL3-CA9p promoter plasmids (wild-type or mutant) in the presence of vector control or flag-tagged BZLF1 (wild-type) or ΔTAD-BZLF1 expressing plasmids. 36 h post-transfection, cells were washed with 1 x PBS and suspended in

100 μl of 1 x passive lysis reagent (PLB). After removing the cell debris, 20 μl of the cell lysate was mixed with 100 μl of Luciferase Assay Reagent (LAR), and luminescence was measured in Synergy H1 microplate reader (BioTek).

## Cell cycle analysis

For cell cycle analyses, ~$10^6$ Jiyoye cells were either left untreated (DMSO control) or treated with lytic cycle inducer 20 ng/ml TPA plus 3 mM sodium butyrate or cell cycle inhibitors–50 μM Mimosine (MedChemExpress) for G0/G1 arrest or 100 μM Nocodazole (MedChemExpress) for mitotic arrest. 24 h post-treatment cells were harvested by centrifugation and washed with ice-cold 1X PBS (Gibco) and fixed with ice-cold 70% ethanol for 30 minutes at 4˚C. Ethanol was added dropwise to the cells with continuous vortexing to avoid any clumping. Cells were then washed twice with ice-cold 1X PBS to remove excess ethanol. To ensure only DNA is stained, 100 μg/ml of RNase-A (Invitrogen) was added to remove RNA from cells. Propidium iodide (Sigma-Aldrich) was added to stain the DNA content at a final concentration of 50 μg/ml and incubated at room temperature for 30 minutes. Ten thousand events were recorded per sample in an S3e Cell Sorter (BIO-RAD). The data were analyzed using FlowJo v10.8.1 software (BD Bioscience).

## Statistical analysis

All the data represented are as the mean values with standard deviation (SD). Statistical significance of differences in the mean values was analyzed using One-Way Anova (Tukey's multiple comparison test) followed by post-Dunnett test or two tailed student's t-test depending on the number of groups. The data was analysed using Microsoft Excel and GraphPad Prism software. -value below 0.05 was considered as significant (*$P < 0.05$; **$P < 0.01$; ***$P < 0.001$; ns, not significant).

## Supporting information

**S1 Fig. Immunofluorescence and qPCR analysis demonstrating infection status of BAC GFP-EBV in resting B-lymphocytes.** (A) Schematic representation of EBV infection (MOI ~10) in primary B-lymphocytes (PBMC) for 2 and 4 days. (B) Immunofluorescence study showing EBV nuclear antigen EBNA2 and GFP expressions. (C) Relative changes of EBNA2 expression in response to EBV infection of primary B-lymphocytes are represented as bar diagrams in comparison to uninfected control samples using B2M as housekeeping gene. Error bars represent standard deviations of triplicate assays of two independent experiments. *, **, *** = p-value < 0.01, 0.005 and 0.001 respectively.
(TIF)

**S2 Fig. Effect of mitogen-induced B-cell activation on CA9 and CA12 expressions.** ~10 x $10^6$ PBMCs isolated from two individual donors were treated with CpG oligo TLR9 ligand ODN2006 for the indicated time points. 2-days and 4-days post-treated cells were harvested, isolated total RNA and subjected for qPCR analyses after converting them to cDNA. The relative changes in transcripts (log10) of the selected genes using the $2^{-\Delta\Delta Ct}$ method are represented as bar diagrams in comparison to untreated PBMC-control using B2M as housekeeping gene. Two independent experiments were carried out in similar settings and results represent as an average value for each transcript. *, **, *** = p-value < 0.01, 0.005 and 0.001 respectively.
(TIF)

**S3 Fig. Expressions of Hifs in EBV transformed cells.** ~5 x $10^6$ PBMCs isolated from two individual donors and two LCL clones (LCL#1 and LCL#89) were harvested, washed with 1 x

PBS, lysed in RIPA buffer and subjected for western blot analyses with the indicated antibodies. GAPDH blot was used as loading control. Representative gel pictures are shown of at least two independent experiments.
(TIF)

**S4 Fig. Reanalyses of microarray and DepMap RNA-Seq data of EBV-negative and positive B-lymphoma cell lines.** Heat map representation of the selected genes of microarray data [36] of EBV infected BL31 cell line.
(TIF)

**S5 Fig. Effect of pan CA inhibitor acetazolamide on cell viability of EBV transformed B-lymphocytes.** LCLs (LCL#1 and LCL#89) were either left untreated (DMSO control) or treated with increasing concentrations (10–1000 μM) of pan carbonic anhydrase inhibitor acetazolamide for 24 h and measured carbonic anhydrase activity, intracellular pH, cell viability and colony formation ability as described in the "Materials and Methods section". Carbonic anhydrase activity and intracellular pH determination assays were performed using kits as per manufacturer's protocols. For cell viability assays, viable cells from each well of 6-well plates were measured by Trypan blue exclusion method using an automated cell counter. For the colony formation assay, 14 days post-treatment colonies on the soft agar were photographed (brightfield) using a Fluorescent Cell Imager. Scale bars, 100 μm. The number of colonies were measured by ImageJ2 software and plotted as bar diagrams. Error bars represent standard deviations of duplicate assays of two independent experiments. *, **, *** = p-value < 0.01, 0.005 and 0.001 respectively.
(TIF)

**S6 Fig. Effect of CA inhibitor U-104 during EBV induced B-cell transformation.** (A-B) As described in Fig 5, PBMCs isolated from donor #3 were infected with GFP-EBV (MOI ~10) in the absence (DMSO control) or in the presence of 0.1 mM carbonic anhydrase inhibitor U-104 for 28 days. (A) At the indicated time points post-infected cells were photographed using a Fluorescent Cell Imager. Scale bars, 100 μm. (B) A fraction of cells at the indicated time points were harvested and subjected to qPCR for CA9 transcript analyses. The relative changes in transcripts (log10) of the selected genes using the $2^{-\Delta\Delta Ct}$ method are represented as bar diagrams in comparison to uninfected PBMC-control using B2M as housekeeping gene. Average values +/- SEM are plotted of two independent experiments performed in similar settings. *, **, *** = p-value < 0.01, 0.005 and 0.001 respectively. (C) Heat map representation of differential gene expression pattern of the indicated viral and CA genes (CA1-CA14) of RNA-Seq data (GSE125974 and DRA011328) of B-cells infected with EBV for the indicated time points. Upregulated: red; downregulated: blue.
(TIF)

**S7 Fig. Role of EBV and its oncoprotein EBNA2 in transcriptional regulation of CA9 and ARHGEF39 genes.** (A) Reanalyses of two ChIP-seq data (GSE176232 and GSE76869) for EBNA2 in GM12878 LCL were reanalysed and displayed using IGV software for CA9 gene locus. (B) Alignment of EBNA2 and RNA-Pol II ChIP-Seq data with ATAC-Seq and ChIA-PET for RNA-Pol II in GM12878 LCL on ARHGEF39 and CA9 genes loci. (C) Reanalyses of RNA-Seq data in PBMCs and GM12878 LCLs for CA9 and ARHGEF39 expressions. (D) PBMCs infected (MOI: ~10) with EBNA2 deleted and wild-type EBV generated from P3HR1 and Jiyoye cells, respectively for the indicated time points were subjected to qPCR analyses for ARHGEF39 expressions. (E) ~10 million P3HR1 and Jiyoye cells were harvested and subjected to qPCR analyses for the indicated viral and cell gene expressions. (D-E) The relative changes in transcripts (log10) of the selected genes using the $2^{-\Delta\Delta Ct}$ method are represented as bar

diagrams in comparison to control samples using B2M as housekeeping gene. Average values +/- SEM are plotted of two independent experiments performed in similar settings. *, **, *** = p-value < 0.01, 0.005 and 0.001 respectively. (F) Dot plot analyses of CA9 and ARHGEF39 expressions in response to EBV infection and EBNA2 expressions of RNA-Seq data among EBV positive and negative B-cell lymphoma cell lines in DepMap portal (https://depmap.org/portal/).
(TIF)

**S8 Fig. ChIP-qPCR analyses of EBNA2 and RBP-Jκ in EBNA2 over-expressed P3HR1 cells.** (A) Cartoon representation of genomic location for three RBP-Jκ binding and non-binding sites on the CA9 promoter region for ChIP-qPCR primer designing. Transcription start site (TSS) is indicated by red arrow. (B-C) ~20 million P3HR1 cells were transiently transfected with control vector or EBNA2 expression construct (pSG5-EBAN2) by electroporation. 48 h post-transfection cells were subjected to ChIP-qPCR analyses using EBNA2 and RBP-Jκ specific antibodies as described in the "Materials and Methods" section. Site 1, 2 and 3 are the binding region and site 4 is non-binding region for EBNA2 and RBP-Jκ as shown in (A). ChIP-qPCR primers were designed by NCBI primer BLAST application. Two independent experiments were carried out in similar settings and results represent as an average value for each genomic segment. *, **, *** = p-value < 0.01, 0.005 and 0.001 respectively.
(TIF)

**S9 Fig. CA inhibitor S4 induces cell apoptosis in P3HR1 cells.** Representative images of flow cytometry analyses showing cell apoptosis in P3HR1 cells after 0.1 mM S4 treatment for 24 h. FITC labelled annexin V binding and PI staining were detected using a BD cell analyzer. Results were analysed using Floreada.io, an online based tool for flow cytometry data analyses.
(TIF)

**S10 Fig. Effect of CA9 knockdown in EBV positive Jiyoye cells.** (A-F) Jiyoye cells stably transduced with lentiviruses expressing either sh-control or sh-RNA (1) against CA9 gene were (A) photographed using a using a Fluorescent Cell Imager and subjected to (B) western blot, (C) qPCR, (D) carbonic anhydrase activity, (E) intracellular pH and (F) cell proliferation analyses as similar to Fig 8O–8T. Error bars represent standard deviations of duplicate assays of two independent experiments. *, **, *** = p-value < 0.01, 0.005 and 0.001 respectively. ns: non-significant.
(TIF)

**S11 Fig. Proteasomal inhibition mediated EBV lytic cycle reactivation promotes CA9 downregulation.** (A-C) LCLs (LCL#1 and LCL#89) either left untreated or treated with 0.5 μM bortezomib for 24 h were subjected to (A) western blot and (B) CA activity analyses. (A) For western blot analyses, GAPDH blot was performed as loading control. Representative gel pictures are shown of at least two independent experiments. (B) Carbonic anhydrase activity was measured using kit as per manufacturer's instructions. Error bars represent standard deviations of duplicate assays of two independent experiments. *, **, *** = p-value < 0.01, 0.005 and 0.001 respectively.
(TIF)

**S12 Fig. BZLF1 expression does not affect CA9 expression in a heterologous system.** ~10 x 10^6 HEK293 cells were transiently transfected with myc-tagged CA9 with or without flag-tagged BZLF1 expression plasmids. 36 h post-transfection cells were harvested and subjected to western blot analyses with the indicated antibodies. GAPDH blot was performed as loading control. The relative intensities (RI) of protein bands shown as bar diagrams were quantified

using the software provided by Odyssey CLx Imaging System. Representative gel pictures are shown of at least two independent experiments.
(TIF)

**S13 Fig. bZIP domain of BZLF1 is responsible for DNA binding activity at CA9 promoter region.** (A) Schematic representation of known structural domains of BZLF1 protein used for cloning in pA3F expression vector.(B-C) ~10 million HEK293 cells were transiently transfected with vector control or expression constructs for wild-type (WT), ΔTAD and ΔbZIP BZLF1 proteins by electroporation. 36 h post-transfection cells were harvested and subjected for either (B) western blot or (C) ChIP-qPCR analyses. (B) For western blot analyses, cells were additionally transfected with GFP expression vector to monitor the transfection efficiency. Western blots were performed with the indicated antibodies, where GAPDH blot was performed as loading control. (C) ChIP-qPCR data showing recruitment of flag-tagged wild-type or ΔTAD BZLF1 proteins on the CA9 promoter region (Site 1 and site 2). ChIP-qPCR primers were designed by NCBI primer BLAST application. Two independent experiments were carried out in similar settings and results represent as an average value for each genomic segment. *, **, *** = p-value < 0.01, 0.005 and 0.001 respectively.
(TIF)

**S14 Fig. Cell cycle arrest does not affect CA9 expression in EBV positive B-lymphocytes.** (A-E) ~10 million Jiyoye cells were treated with (A) lytic cycle inducer 20 ng/ml TPA plus 3 mM sodium butyrate (NaBu) or (B-E) cell cycle inhibitors–(B-C) 50 μM Mimosine for G0/G1 arrest or (D-E) 100 μM Nocodazole for mitotic arrest. 24 h post-treatment cells were harvested and subjected to either (A-B and D) flow cytometry or (C and E) western blot analyses. (A-B and D) Representative cell cycle distribution of propidium iodide stained Jiyoye cells using an S3e Cell Sorter. (C-E) Western blots were performed with the indicated antibodies, where GAPDH blot was performed as loading control.
(TIF)

**S1 Table. Ki values of carbonic anhydrase (CA) inhibitors.**
(DOCX)

**S2 Table. Previously published datasets used in this study.**
(DOCX)

**S3 Table. Oligo sequences used for qPCR, semi-qPCR, ChIP-qPCR, Sh-RNAs and cloning of cDNA and promoter regions.**
(DOCX)

## Acknowledgments

We sincerely thank to Erle S Robertson (Perelman School of Medicine, University of Pennsylavania, USA), Rupak Dutta (Indian Institute of Science Education and Research, Kolkata, India), Debanjan Mukhopadhyay and Somsubhra Nath (Presidency University, Kolkata), William S Sly (Saint Louis University School of Medicine), S. Diane Hayward (Johns Hopkins University School of Medicine), Didier Trono (Swiss Federal Institute of Technology in Lausanne), Simon Davis (University of Oxford) and National Centre for Cell Science (NCCS), Dept. of Biotechnology (DBT), Govt. of India for providing reagents, plasmids, and cell lines. We thank Piyali Mukherjee (Presidency University, Kolkata, India) for careful review of the manuscript.

## Author Contributions

**Conceptualization:** Abhik Saha.

**Data curation:** Samaresh Malik, Joyanta Biswas, Purandar Sarkar, Subhadeep Nag, Chandrima Gain, Shatadru Ghosh Roy, Abhik Saha.

**Formal analysis:** Samaresh Malik, Joyanta Biswas, Shatadru Ghosh Roy, Abhik Saha.

**Funding acquisition:** Abhik Saha.

**Investigation:** Samaresh Malik, Joyanta Biswas, Purandar Sarkar, Subhadeep Nag, Abhik Saha.

**Methodology:** Samaresh Malik, Joyanta Biswas, Bireswar Bhattacharya, Dipanjan Ghosh, Abhik Saha.

**Project administration:** Abhik Saha.

**Resources:** Dipanjan Ghosh, Abhik Saha.

**Supervision:** Abhik Saha.

**Validation:** Samaresh Malik, Joyanta Biswas.

**Visualization:** Samaresh Malik, Joyanta Biswas.

**Writing – original draft:** Abhik Saha.

**Writing – review & editing:** Abhik Saha.

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
