## [Decision Letter · Decision Letter 0]

1 Mar 2024

Dear Dr. Saha,

We are pleased to inform you that your manuscript 'Differential Carbonic Anhydrase Activities Control EBV-Induced B-Cell Transformation and Lytic Cycle Reactivation' has been provisionally accepted for publication in PLOS Pathogens.

Best regards,

Sumita Bhaduri-McIntosh, M.D., Ph.D.

Academic Editor

PLOS Pathogens

Patrick Hearing

Section Editor

PLOS Pathogens

Michael Malim

Editor-in-Chief

PLOS Pathogens

orcid.org/0000-0002-7699-2064

Reviewer Comments (if any, and for reference):

Reviewer's Responses to Questions

**Part I - Summary**

Reviewer #1: The authors demonstrate a critical role for Carbonic Annhydrase in control of EBV immortalization of B-cells. Some of the mechanisms remain unconventional, like the repression of CA9 by BZLF1 activation domain, but the authors provide sufficient data to support these claims. They have addressed my major concerns in the revised version and rebuttal.

Reviewer #2: In this revised manuscript, the authors have presented novel additional findings that effectively address prior concerns. Consequently, the significance of the results has substantially increased, better supporting the authors' conclusions. However, the precise functional role of repressing CA9 by BZLF1 during lytic reactivation remains somewhat unclear, and this aspect should be acknowledged as a limitation in the discussion.

**Part II – Major Issues: Key Experiments Required for Acceptance**

Reviewer #1: (No Response)

Reviewer #2: (No Response)

**Part III – Minor Issues: Editorial and Data Presentation Modifications**

Reviewer #1: (No Response)

Reviewer #2: (No Response)

PLOS authors have the option to publish the peer review history of their article (what does this mean?). If published, this will include your full peer review and any attached files.

Reviewer #1: No

Reviewer #2: No

---

## [Editor Report · Acceptance letter]

21 Mar 2024

Dear Dr. Saha,

We are delighted to inform you that your manuscript, "Differential Carbonic Anhydrase Activities Control EBV-Induced B-Cell Transformation and Lytic Cycle Reactivation," has been formally accepted for publication in PLOS Pathogens.

Best regards,

Michael Malim

Editor-in-Chief

PLOS Pathogens

orcid.org/0000-0002-7699-2064